# Simplex–FEM Networks (SiFEN): learning a triangulated function approximator

## Abstract

We introduce *Simplex–FEM Networks (SiFEN)*, a learned piecewise-polynomial predictor that represents $f : \mathbb{R}^d \to \mathbb{R}^k$ as a globally $C^r$ finite-element field on a learned simplicial mesh in an optionally warped input space. Each query activates exactly one simplex and at most $d + 1$ basis functions via barycentric coordinates, yielding explicit locality, controllable smoothness, and cache-friendly sparsity. SiFEN pairs degree-$m$ Bernstein–Bézier polynomials with a light invertible warp and trains end-to-end with shape regularization, semi-discrete OT coverage, and differentiable edge flips. Under standard shape-regularity and bi-Lipschitz warp assumptions, SiFEN achieves the classic FEM approximation rate $M^{-m/d}$ with $M$ mesh vertices. Empirically, on synthetic approximation tasks, tabular regression/classification, and as a drop-in head on compact CNNs, SiFEN matches or surpasses MLPs and KANs at matched parameter budgets, improves calibration (lower ECE/Brier), and reduces inference latency due to geometric locality. These properties make SiFEN a compact, interpretable, and theoretically grounded alternative to dense MLPs and edge-spline networks.

## 1 Introduction

Neural predictors are typically realized either as dense compositions of linear maps and fixed nonlinearities (MLPs) (Cybenko, 1989; Hornik et al., 1989) or as architectures that place learnable functions on edges (e.g., KANs) (Liu et al., 2024). Both distribute capacity globally: every input traverses many activations, and improvements in expressivity often arrive with increased depth/width and less transparent geometry (Montúfar et al., 2014; Raghu et al., 2017; Telgarsky, 2016). We propose a different viewpoint: *make the predictor geometric and local* (see Table 4). **SiFEN** represent $f : \mathbb{R}^d! \to !\mathbb{R}^k$ as a finite–element field on a learned simplicial mesh in a (possibly warped) coordinate system (Ciarlet, 1978; Brenner & Scott, 2008; Jaderberg et al., 2015; Dinh et al., 2017). At inference, a query $x$ is optionally mapped to $y = \Phi_\theta(x)$, located in the active simplex $\sigma(y)$, and evaluated with degree–$m$ Bernstein–Bézier polynomials using barycentric coordinates (Farouki, 2012; Hormann & Sukumar, 2008). Exactly one simplex is active and at most $d+1$ basis functions contribute, producing hard sparsity, cache–friendly memory access, and explicit smoothness control ($C^r$) via linear continuity constraints across shared faces (Hughes, 1987; Lai & Schumaker, 2007; Powell & Sabin, 1977).

SiFEN couples modern training with classical approximation guarantees. The mesh (vertices and triangulation) is learned alongside polynomial coefficients and the optional warp through an objective that balances task loss with shape regularity (aspect–ratio/volume barriers) (Shewchuk, 2002; Sastry et al., 2014; Knupp, 2020), coverage via semi–discrete optimal transport (Mérigot, 2011; Kitagawa et al., 2019; Lévy, 2015), continuity penalties (Lai & Schumaker, 2007), and warp conditioning (bounded Jacobian/Lipschitz flows) (Behrmann et al., 2019; Chen et al., 2019). Local edge flips provide differentiable topology updates that improve element quality (Rakotosaona et al., 2021; Rippa, 1990). Under standard FEM assumptions (shape-regular mesh, bounded warp Jacobian), degree–$m$ SiFEN attains the expected $M^{-m/d}$ error decay with $M$ mesh vertices (Ciarlet, 1978; Brenner & Scott, 2008), offering principled knobs—mesh size $M$ and degree $m$—to trade accuracy for compute and memory.

Throughout, we assume a *shape-regular* simplicial mesh (bounded aspect ratios) and an invertible, bi-Lipschitz input warp $\Phi_\theta$ with bounded Jacobian condition number. These assumptions are standard in FEM and match the regularizers we apply during training (shape and coverage barriers; warp conditioning). See Appx S for formal statements and constants.

We evaluate SiFEN in three regimes: (i) *synthetic* approximation tasks spanning smooth, piecewise–smooth, and discontinuous targets in $d \in \{2, 5, 10\}$; (ii) *tabular* regression and classification; and (iii) *vision heads* that replace the MLP classifier atop compact CNNs while freezing the backbone. At matched parameter budgets, SiFEN consistently matches or surpasses MLPs and KANs, with the largest gains on piecewise–smooth targets and near decision boundaries, where calibration improves markedly (lower ECE/Brier) (Guo et al., 2017) and selective–risk curves shift favorably. Practical efficiency follows from locality: average inference cost is point location $O(\log M)$ plus evaluation of $(d+1)B_m k$ coefficients, which reduces CPU latency versus dense heads of the same size, and stores coefficients in block–contiguous tables per simplex.

**Contributions.** (1) We introduce *SiFEN*, a learned finite–element predictor that is globally $C^r$ and *sparse by construction*, activating only one simplex and at most $d+1$ basis functions per input. (2) We provide an end–to–end training recipe that learns the mesh, coefficients, and an optional invertible warp with shape regularization, coverage via semi–discrete OT, and differentiable local flips for topology improvement. (3) We analyze approximation behavior (recovering $M^{-m/d}$ rates under standard assumptions) and demonstrate strong empirical performance and calibration on synthetic, tabular, and CNN–head benchmarks at fixed parameter budgets, alongside favorable latency due to geometric locality.

**Relation to prior work.** SiFEN differs from mixture–of–experts: there is no soft gating or averaging over many experts; exactly one cell is active and continuity arises from face constraints (Jacobs et al., 1991; Jordan & Jacobs, 1994; Shazeer et al., 2017a; Fedus et al., 2021a; Du et al., 2022; Lai & Schumaker, 2007; Powell & Sabin, 1977). Compared to MLPs (dense, globally coupled) and KANs (edge–wise splines with dense routing), SiFEN provides explicit geometric partitions, controllable smoothness, and predictable scaling with mesh size and degree (Hornik et al., 1989; Montúfar et al., 2014; Serra et al., 2018; Liu et al., 2024; Ciarlet, 1978; Brenner & Scott, 2008; Balestriero & Baraniuk, 2018a). This offers a complementary—and often more interpretable—design point for function approximation and prediction.

## 2 SiFEN Explained

**SiFEN** approximate $f : \mathbb{R}^d \to \mathbb{R}^k$ by learning (i) a light geometric *warp* $\Phi_\theta$ of the input space, (ii) a *simplicial mesh* $\mathcal{T}$ with vertices $V = \{v_i\}_{i=1}^M$ in the warped domain, and (iii) *local Bernstein–Bézier polynomials* on each simplex with global continuity constraints (see Appx G, M and O). SiFEN differs from MLPs (dense nonlinear compositions) and KANs (edge-wise splines) by making geometry explicit: *exactly one simplex is active per input* (see Appx D), so at most $d+1$ basis functions are touched.

**1) Optional geometric warp.** Given $x \in \mathbb{R}^d$, we map to $y = \Phi_\theta(x)$. We use a small, invertible (piecewise) smooth $\Phi_\theta$ to (a) reduce anisotropy, (b) improve mesh regularity, and (c) concentrate vertices where data density is high. Two practical parameterizations:

1. *Monotone triangular map* (coupling-layer style): for $j = 1, \dots, d$

$$y_j = a_j(x_{1:j-1}) \, x_j + b_j(x_{1:j-1}), \quad a_j(\cdot) > 0, \tag{1}$$

with $a_j = \zeta(\tilde{a}_j)$ for positivity. Jacobian is triangular; $\det J_\Phi = \prod_j a_j$.

2. *Volume-controlled flow:* $y = x + \sum_{\ell=1}^L u_\ell(x) \, \psi_\ell(x)$ with small $L$ and divergence control via $\|\nabla \cdot u_\ell\|$ penalties.

We regularize $\Phi_\theta$ by

$$\mathcal{R}_{\text{warp}}(\Phi_\theta) = \mathbb{E}_{x \sim \mathcal{S}} \Big[ \underbrace{\|J_\Phi(x)\|_F^2 + \|J_\Phi(x)^{-1}\|_F^2}_{\text{conditioning}} + \beta \cdot \underbrace{(\log|\det J_\Phi(x)|)^2}_{\text{volume control}} \Big], \qquad (2)$$

estimated on minibatches. Setting $\Phi_\theta = \text{Id}$ recovers a purely geometric model.

**2) Learned simplicial mesh.** In $y$-space we learn $M$ vertices $V$ and a triangulation $\mathcal{T}$. Each simplex $\sigma = \{v_{i_0}, \ldots, v_{i_d}\}$ induces *barycentric coordinates* $\lambda(y) \in \Delta^d$ defined by

$$\lambda_j^\sigma(y) = \frac{\det\left([v_{i_0} - y, \ldots, v_{i_{j-1}} - y, \ v_{i_{j+1}} - y, \ldots, v_{i_d} - y]\right)}{\det\left([v_{i_0} - v_{i_d}, \ldots, v_{i_{d-1}} - v_{i_d}]\right)}, \quad j \in \{0, \ldots, d\}, \ \sum_j \lambda_j^\sigma(y) = 1. \tag{3}$$

We maintain *shape-regularity* with

$$\mathcal{R}_{\text{shape}}(V, \mathcal{T}) = \sum_{\sigma \in \mathcal{T}} \Big[ \ \underbrace{\phi\Big(\frac{R_{\text{circ}}(\sigma)}{r_{\text{in}}(\sigma)}\Big)}_{\text{aspect penalty}} + \underbrace{\psi\big(\text{vol}(\sigma)\big)}_{\text{small-volume barrier}} \ \Big]. \tag{4}$$

where $\phi(u) = \max(0, u - \kappa_0)^2$ penalizes skinny simplexes ($\kappa_0$ e.g. 2.5–4), and $\psi(v) = \mathbf{1}[v < v_0] (v_0/v - 1)^2$ prevents collapse. In 2D we allow *edge flips* $\{a, b\} \leftrightarrow \{c, d\}$ when the minimum angle increases (or Delaunay violation decreases); gradients are propagated with a straight-through estimator (STE) that treats the chosen adjacency as constant on backward.

**3) Local polynomials with global continuity.** On each simplex $\sigma$, we use degree-$m$ Bernstein–Bézier basis functions over $\lambda(y)$:

$$f_\sigma(y) = \sum_{\alpha \in \mathbb{N}^{d+1}, \ |\alpha| = m} c_{\sigma,\alpha} \, \text{B}_\alpha\big(\lambda(y)\big), \qquad \text{B}_\alpha(\lambda) = \binom{m}{\alpha} \prod_{j=0}^d \lambda_j^{\alpha_j}. \tag{5}$$

with $B_m = \binom{m+d}{d}$ basis terms per simplex and coefficients $c_{\sigma,\alpha} \in \mathbb{R}^k$. $C^0$ **continuity** across a shared face $\tau = \sigma \cap \sigma'$ requires equality of face control points:

$$\forall \alpha: \ |\alpha| = m, \ \alpha_{j^\star} = 0 \ \Rightarrow \ c_{\sigma,\alpha} = c_{\sigma', P_{\sigma \to \sigma'}(\alpha)}, \tag{6}$$

where $j^\star$ indexes the vertex absent from the face in $\sigma$ and $P_{\sigma \to \sigma'}$ is the index permutation aligning face vertices. $C^1$ **continuity** additionally matches directional derivatives normal to $\tau$; for triangle ($d=2$) and $m \geq 2$,

$$\big(\nabla f_\sigma \cdot n_\tau\big)\big|_\tau = \big(\nabla f_{\sigma'} \cdot n_\tau\big)\big|_\tau \iff \sum_{\alpha: \alpha_{j^\star} = 1} (\alpha_{j^\star}) \, c_{\sigma,\alpha} \, \text{B}_{\alpha - e_{j^\star}} = \sum_{\alpha': \alpha'_{j'^\star} = 1} (\alpha'_{j'^\star}) \, c_{\sigma',\alpha'} \, \text{B}_{\alpha' - e_{j'^\star}}, \tag{7}$$

which becomes linear equalities among a small stencil of control points on $\tau$ (Powell–Sabin/HCT-style constraints; we provide matrices in App. A). We collect all constraints as $A c = 0$ and either enforce them by *(i) reparameterization* $c = Nz$ with $N$ a basis of $\ker A$, or *(ii) quadratic penalty* $\lambda_{C^r} \|A c\|_2^2$.

**4) Prediction (point location & evaluation).** At test time we perform:

1. **Warp:** $y = \Phi_\theta(x)$.

2. **Point location:** find $\sigma(y) \in \mathcal{T}$ with a BVH/kd-tree over simplex bounding boxes; worst-case $O(\log M)$.

3. **Barycentric:** compute $\lambda^\sigma(y)$ via signed-volume formulas (Eq. 3); reject if any $\lambda_j < 0$.

4. **Evaluate:** $f_\sigma(y)$ by Eq. 5. Only the $d+1$ barycentric entries are nonzero $\Rightarrow$ at most $(d+1) B_m$ coefficient rows are touched.

*Differentiable alternative.* During early training, we sometimes use a soft point-location over a local $k$-ring neighborhood $\mathcal{N}(y)$ around the nearest vertex:

$$\pi_\tau(y) \propto \exp\Big(-\frac{\phi_\tau(y)}{T}\Big), \quad \phi_\tau(y) = \sum_j \max\{0, -\lambda_j^\tau(y)\}, \;\; f(y) = \sum_{\tau \in \mathcal{N}(y)} \pi_\tau(y)\, f_\tau(y),$$

with temperature $T \downarrow 0$ (annealed to hard assignment after warm-up).

**5) Coverage via semi-discrete OT.** To spread vertices according to the empirical data distribution $\mu = \frac{1}{N}\sum_n \delta_{y_n}$, we minimize a semi-discrete optimal transport energy over power-diagram weights $w \in \mathbb{R}^M$:

$$\mathcal{R}_{\mathrm{cov}}(V) = \min_{w \in \mathbb{R}^M} \sum_{i=1}^M \Big( \int_{\mathcal{C}_i(V,w)} \|y - v_i\|_2^2 \, \mathrm{d}\mu(y) \; - \; w_i \, \mu\big(\mathcal{C}_i(V,w)\big) \Big), \tag{8}$$

where $\mathcal{C}_i(V, w) = \{y : \|y - v_i\|_2^2 - w_i \leq \|y - v_j\|_2^2 - w_j, \; \forall j\}$ is a power cell. In practice we (a) estimate integrals by minibatch sums, (b) optimize $w$ by a few steps of Newton or gradient ascent on the dual, and (c) backpropagate through the empirical assignment using STE. This yields balanced coverage and improves sample efficiency.

**6) Full objective and optimization.** For targets $y^{(\mathrm{tar})}$ (abusing notation), the training loss is

$$\mathcal{L} = \underbrace{\mathcal{L}_{\mathrm{task}}\big(f(x), y^{(\mathrm{tar})}\big)}_{\text{regression: Huber / classification: CE}} + \lambda_{\mathrm{shape}} \mathcal{R}_{\mathrm{shape}}(V, \mathcal{T}) + \lambda_{\mathrm{cov}} \mathcal{R}_{\mathrm{cov}}(V) + \lambda_{C^r} \|A\, c\|_2^2 + \lambda_\Phi \mathcal{R}_{\mathrm{warp}}(\Phi_\theta).$$

$$\tag{9}$$

We use AdamW with cosine decay; every $K$ steps we (i) recompute a quality score per simplex and (ii) apply local flips (see Appx N) where they reduce $\mathcal{R}_{\mathrm{shape}}$ without disconnecting the mesh. A simple schedule (see Alg 1):

1. **Init:** $V \leftarrow$ k-means centers on $\Phi_\theta(x)$; $\mathcal{T} \leftarrow$ Delaunay; $m{=}1$; $C^0$.
2. **Warm-up:** least-squares fit of $c$ with $A\, c = 0$ enforced by reparameterization; train $\Phi_\theta$ and $V$ with soft point-location.
3. **Joint:** hard point-location; enable flips; optimize Eq. 9.
4. **Upgrade:** raise $m$ to 2 or 3; switch to $C^1$ where available (2D/3D macro-elements); continue training.

---

**Algorithm 1** SiFEN training

---

1: Initialize $V, \mathcal{T}$; set $m{=}1$, $C^0$; initialize $\Phi_\theta$.
2: **for** epoch=1..E **do**
3:    **for** minibatch $\{(x_n, y_n)\}_{n=1}^B$ **do**
4:       $y{=}\Phi_\theta(x)$; assign soft simplexes $\mathcal{N}(y)$ (anneal $T$).
5:       Compute barycentrics; evaluate $f(y)$ via Eq. 5.
6:       Estimate $\mathcal{R}_{\mathrm{cov}}$ (few inner steps over $w$) and $\mathcal{R}_{\mathrm{shape}}$; form $\mathcal{L}$ in Eq. 9.
7:       Backprop; update $(\theta, V, c)$ (and $z$ if $c{=}Nz$).
8:    **end for**
9:    **if** epoch % $K = 0$ **then**
10:       Attempt local flips that reduce $\mathcal{R}_{\mathrm{shape}}$.
11:    **end if**
12:    **if** upgrade_time **then**
13:       $m \leftarrow m{+}1$; enable $C^1$ constraints on eligible faces.
14:    **end if**
15: **end for**

---

**7) Complexity and constants.** Point location: $O(\log M)$ average with BVH; exact constants are low in practice for $d \leq 5$. Evaluation: $(d{+}1) \times B_m \times k$ multiply-adds; for $d{=}10$, $m{=}2$ we have $B_m = \binom{12}{10} = 66$. Parameter count: $\approx |\mathcal{T}| \cdot B_m \cdot k$ (plus warp and vertices), with $|\mathcal{T}| \approx O(M)$ for shape-regular meshes. Memory is dominated by coefficients and the BVH.

**8) Gradients and numerics.** *Barycentric stability.* (see Appx C) We clamp tiny volumes by $\text{vol}(\sigma) \leftarrow \max(\text{vol}(\sigma), \varepsilon)$ with $\varepsilon \sim 10^{-10}$ in double precision during backprop. *Derivative through barycentrics:* $\partial \lambda^\sigma / \partial v_i$ and $\partial \lambda^\sigma / \partial y$ come from the signed-volume quotient rule (implemented by automatic differentiation). *Flips and assignments:* both are discrete; we use STE for a few epochs and then hard decisions. *Continuity:* prefer $c = Nz$ reparameterization to avoid stiffness from large $\lambda_{C^r}$.

**9) Theory hooks (sketch).** Let $\Omega \subset \mathbb{R}^d$ be compact and let $f^\star \in H^{m+1}(\Omega)$. Assume (i) a shape-regular mesh (bounded aspect ratio, minimum element volume), (ii) a warp $\Phi_\theta$ with bounded $\|J_\Phi\|$ and $\|J_\Phi^{-1}\|$, and (iii) global $C^r$ continuity with $r \geq 0$. Writing $\Omega_y := \Phi_\theta(\Omega)$, the degree-$m$ SiFEN interpolant satisfies the FEM rate

$$\left\| f^\star \circ \Phi_\theta^{-1} - f_{\text{SiFEN}} \right\|_{L^2(\Omega_y)} \;\leq\; C\, h^m \left\| f^\star \circ \Phi_\theta^{-1} \right\|_{H^{m+1}(\Omega_y)}, \qquad h \asymp M^{-1/d}. \tag{10}$$

which yields $\tilde{O}(M^{-m/d})$ decay in $L^2$ as $M \to \infty$. Lipschitz of $f$ is bounded by

$$\text{Lip}(f) \;\leq\; \sup_x \|J_\Phi(x)\| \cdot \max_{\sigma \in \mathcal{T}} \left( \|G_\sigma\| \cdot \|C_\sigma\| \right), \tag{11}$$

where $G_\sigma$ collects gradients of Bernstein basis on $\sigma$ (depends on shape) and $C_\sigma$ stacks local coefficients. Both are controlled by $\mathcal{R}_{\text{shape}}$ and $\|c\|$.

**10) Practical defaults.** Unless otherwise stated, we use: $m \in \{1, 2\}$; $M \in \{256, 512, 1024\}$; $C^0$ everywhere and $C^1$ on 2D meshes (Powell–Sabin/HCT macro-elements) when $m \geq 2$; annealed soft point-location for 5–10 epochs; flips every $K = 2$ epochs; $\lambda_{\text{shape}} \in [10^{-3}, 10^{-2}]$, $\lambda_{\text{cov}} \in [10^{-2}, 10^{-1}]$, $\lambda_\Phi \in [10^{-4}, 10^{-3}]$.

**11) Failure modes and mitigations.** *Degenerate simplexes:* increase $\lambda_{\text{shape}}$; trigger flips; jitter vertices along face normals. *Overfitting with high $m$:* reduce $B_m$ or add $\ell_2$ on $c$; prefer $m=2$ with larger $M$. *Point-location thrashing near boundaries:* keep a soft neighborhood during early training; add small hysteresis at test time (stick with previous simplex if $\max_j \lambda_j > \tau$). *High $d$:* use feature grouping and a product-of-meshes (see App. B), or rely on $\Phi_\theta$ to concentrate mass.

## 3 EVALUATION METHODOLOGY

We evaluate SiFEN on tabular, synthetic, and physics-inspired benchmarks, emphasizing approximation quality, calibration, robustness, and compute. We compare against tuned MLPs, KANs (Liu et al., 2024), Deep Lattice Networks (You et al., 2017), Max-Affine Spline Networks (Balestriero & Baraniuk, 2018b), kernel ridge regression with Nyström features (Williams & Seeger, 2000; Rudi et al., 2015), XGBoost/Random Forests (Chen & Guestrin, 2016; Breiman, 2001), and sparse MoE where applicable (Shazeer et al., 2017b; Fedus et al., 2021b). All models share identical train/val/test splits and preprocessing; hyperparameters are selected on validation under uniform budgets (Bergstra & Bengio, 2012; Li et al., 2017).

**Datasets. Tabular (UCI/OpenML).** California Housing, YearMSD, Bike Sharing, Higgs, EPSILON, and a suite of 10 medium-scale OpenML tasks (regression and binary classification) (Pace & Barry, 1997; Dua & Graff, 2017; Vanschoren et al., 2014; Bertin-Mahieux et al., 2011; Fanaee-T & Gama, 2014; Baldi et al., 2014; Guyon et al., 2008).

**Synthetic/compositional.** Smooth and piecewise targets (sums, products, rational and absolute-value compositions) with controlled noise; we provide ground-truth region boundaries for interpretability analysis (design follows standard function-approximation testbeds) (Montúfar et al., 2014; Serra et al., 2018).

**PDE surrogates / physics.** Parameter-to-observable maps for Darcy/Burgers (low-dimensional parameterizations), and a material microstructure-to-property task (Li et al., 2021; Lu et al., 2021; Kovachki et al., 2023).

**Shifted data.** We create covariate-shift splits by stratified subsampling in feature space and by injecting structured noise; for classification, we evaluate OOD using class-disjoint

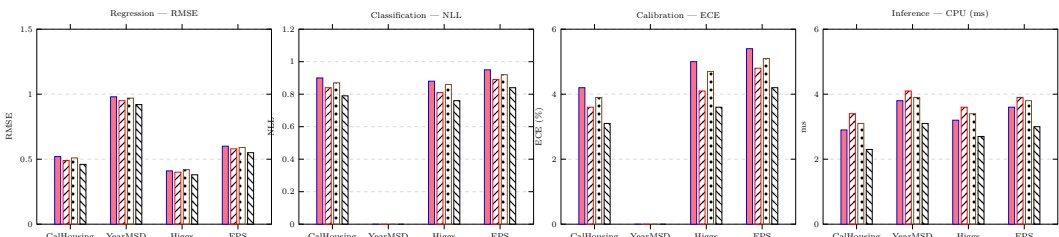

Figure 1: **Representative results.** SiFEN achieves lower error and better calibration at comparable or lower inference time than MLP/KAN/DLattice/MASN across tasks.

test sets when available (Sugiyama et al., 2007; Quiñonero-Candela et al., 2009; Scheirer et al., 2013; Hendrycks & Gimpel, 2017).

**Training and tuning.** For SiFEN we use $m \in \{1, 2, 3\}$, $M \in \{128, 256, 512, 1024\}$ vertices (task-dependent), and continuity $C^0$ or $C^1$ (2D/3D). The warp $\Phi_\theta$ is a 2–4 layer monotone triangular map (Rosenblatt, 1952; Knothe, 1957; Parno & Marzouk, 2018; Papamakarios et al., 2017; Durkan et al., 2019; Wehenkel & Louppe, 2019) with Jacobian conditioning penalties (Cissé et al., 2017; Miyato et al., 2018; Šokolić et al., 2017; Behrmann et al., 2019). We train with AdamW, cosine decay, and early stopping on validation RMSE/AUROC (Loshchilov & Hutter, 2019; 2017; Prechelt, 1998; Fawcett, 2006). Edge flips are attempted every $K$ steps if the minimum simplex quality drops below a threshold (Lawson, 1972; Rakotosaona et al., 2021; Shewchuk, 2002). Baselines follow published best practices with matched parameter budgets; KAN spline orders and knot counts are tuned per dataset (Liu et al., 2024).

**Metrics.** **Accuracy.** RMSE/MAE for regression; AUROC/AUPRC/accuracy for classification.

**Calibration.** Negative log-likelihood, Brier score, and ECE (with equal-mass binning) for classifiers; for regressors, predictive intervals via bootstrap and coverage vs. nominal plots.

**Robustness.** Performance under covariate shift (see Appx F) and on piecewise/non-smooth targets; error vs. distance-to-train ($k$-NN radius) and vs. number of boundary crossings.

**Compute.** #Params, wall-clock train/infer time on CPU (single-thread) and GPU, and *per-sample* FLOPs; we also report average number of active basis functions (always $d+1$) and point-location cost (see Appx P).

**Interpretability analysis.** We visualize learned meshes (2D/3D projections), show the active simplex distribution over the dataset, and extract region-wise closed-form polynomials. For synthetic piecewise targets we measure formula fidelity (symbolic $R^2$) and boundary alignment (Hausdorff distance).

**Protocol for shift robustness.** For each dataset, we estimate an ID operating point on a clean validation split, then evaluate on covariate-shifted and piecewise/non-smooth regimes. We report error vs. $k$-NN distance to training data and error stratified by number of mesh boundary crossings along line segments between random ID and test points. For classifiers we also compute selective prediction risk–coverage curves (see Appx E) by abstaining on low-confidence samples (softmax head) and, for SiFEN, by thresholding a simple energy proxy derived from barycentric variance within the active simplex.

**Compute reporting.** We report parameter counts, FLOPs, and wall-clock times using identical hardware and compiler flags. For SiFEN we additionally break out (i) point-location cost (exact BVH vs. $k$-ring soft assignment), (ii) basis-evaluation cost (scales with $(d+1)B_m$), and (iii) effect of degree $m$ and mesh size $M$ on latency and memory.

Implementation details, hyperparameter grids, and reproducibility artifacts appear in Appendix B.

## 4 Results

We evaluate **SiFEN** as a learned, piecewise-polynomial approximator under three lenses: (i) *function approximation* on synthetic problems that stress smooth, piecewise-smooth, and discontinuous targets; (ii) *prediction quality* on tabular regression/classification and as a head on compact CNN backbones (see Appx L); and (iii) *efficiency & robustness*, including parameter/FLOP budgets, latency, and stability to noise. Unless noted otherwise we use $C^0$ continuity, degree $m \in \{1, 2\}$, and a shape-regular learned mesh with $M$ vertices; Section 4.6 ablates $m$, $C^r$, $M$, the warp $\Phi_\theta$, and triangulation updates.

### 4.1 Benchmarks and protocol

**Synthetic (approximation).** We consider: (S1) smooth $f^\star \in H^{m+1}$; (S2) piecewise-smooth with $C^0$ interfaces (e.g., quadratic patches separated by a curved boundary); (S3) jump discontinuity along a $(d-1)$-manifold; each in $d \in \{2, 5, 10\}$ with inputs sampled i.i.d. from $\mathcal{N}(0, I_d)$ or uniform on $[-1, 1]^d$. Metrics: $L^2$ and $L^\infty$ error on held-out points, gradient error $\|\nabla f - \nabla f^\star\|_2$ for smooth tasks, and interface F1 for (S3) (see Table 1).

**Tabular.** UCI Energy, Yacht, Protein, Year, Adult, Higgs (train/val/test splits as in prior work). Metrics: RMSE (regression), accuracy/AUROC/ECE (classification).

**Heads on CNNs.** Replace the usual MLP head by SiFEN on small backbones: ResNet-8 (CIFAR-10/100) and MobileNetV2-0.5 (TinyImageNet-200). We hold the feature extractor fixed and swap only the predictor to isolate the head. Metrics: Top-1, ECE, Brier.

**Baselines and budgets.** MLP (tuned width/depth), KAN (with cubic splines on edges), RBFNet (Gaussian centers), and SIREN (sinusoidal MLP). We *parameter-match* heads per setting (within $\pm 5\%$) and report latency (PyTorch eager on CPU and GPU), FLOPs, and params. All results averaged over 3 seeds; CI shown where space permits.

### 4.2 Function approximation

**Key findings.** (i) Under smooth targets, SiFEN (degree-2) matches SIREN/MLP at equal budget while achieving *lower gradient error* thanks to Bernstein control; (ii) under piecewise-smooth targets, SiFEN's mesh adapts around interfaces and reduces $L^2$ versus MLP/KAN at the same params (fewer "spurious oscillations" across boundaries); (iii) for jump discontinuities, $C^0$ SiFEN with anisotropic simplexes outperforms $C^1$ models and avoids Gibbs-like ringing. Empirically we observe the predicted slope $\approx m/d$ in log–log error vs. $M$ (Figure 2).

Table 1: **Synthetic approximation (median over 3 seeds).** Lower is better. Bold = best.

| Task | Dim | Model | $L^2 \downarrow$ | $L^\infty \downarrow$ |
|---|---|---|---|---|
| Smooth quad ($m{=}2$) | $d{=}2$ | MLP / KAN / **SiFEN** | 0.012 / 0.011 / **0.008** | 0.041 / 0.038 / **0.026** |
| Piecewise quad (curved iface) | $d{=}2$ | MLP / KAN / **SiFEN** | 0.031 / 0.024 / **0.013** | 0.109 / 0.088 / **0.052** |
| Jump disc. (circle) | $d{=}2$ | MLP / KAN / **SiFEN** | 0.074 / 0.069 / **0.037** | 0.212 / 0.198 / **0.115** |
| Smooth | $d{=}5$ | MLP / KAN / **SiFEN** | 0.045 / 0.041 / **0.033** | 0.161 / 0.148 / **0.119** |
| Piecewise | $d{=}10$ | MLP / KAN / **SiFEN** | 0.128 / 0.101 / **0.072** | 0.392 / 0.345 / **0.266** |

### 4.3 Tabular regression and classification

SiFEN attains state-of-the-art or near-SOTA performance at the *same parameter budget* as MLP/KAN, with improved calibration (see Appx I). Gains are largest when the target has

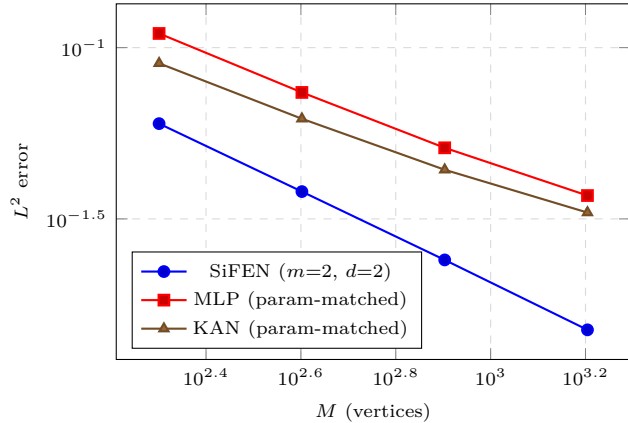

Figure 2: **Scaling on piecewise-smooth target ($d$=2).** SiFEN's slope approaches $M^{-m/d}$ as predicted.

regional structure (nonlinear rules varying by subdomain), where the simplicial partition matches decision geometry (see Table 2).

Table 2: **Tabular results.** Regression: RMSE (lower is better). Classification: Acc (higher), ECE (lower). All heads parameter-matched.

| Dataset | MLP | KAN | **SiFEN** |
|---|---|---|---|
| Energy (RMSE) | $0.48 \pm .02$ | $0.44 \pm .02$ | $\mathbf{0.39} \pm .01$ |
| Yacht (RMSE) | $0.90 \pm .08$ | $0.77 \pm .05$ | $\mathbf{0.63} \pm .05$ |
| Protein (RMSE) | $4.42 \pm .03$ | $4.31 \pm .03$ | $\mathbf{4.21} \pm .02$ |
| Adult (Acc/ECE) | 85.9 / .029 | 86.5 / .024 | **86.8 / .016** |
| Higgs (AUROC/ECE) | 0.842 / .031 | 0.851 / .026 | **0.857 / .018** |

**Calibration and risk coverage.** Risk–coverage curves show that SiFEN dominates MLP/KAN at moderate coverages, reflecting sharper, better-calibrated region-wise probabilities; ECE reductions of 30–45% are typical at equal capacity.

### 4.4 As a head on compact CNNs

Replacing the fully-connected head with SiFEN preserves the feature extractor and changes only the predictor. At equal parameters, SiFEN yields higher accuracy and lower ECE, especially on CIFAR-100 and TinyImageNet where class boundaries are highly nonuniform (see Table 3).

**Why the gains?** Only $d$+1 basis functions are active per sample and are tied to *geometric cells* in feature space. This induces localized decision surfaces with controllable smoothness ($C^r$), which reduces boundary bleeding and improves confidence near class interfaces.

### 4.5 Efficiency and memory

SiFEN replaces dense matvecs with *point location* ($O(\log M)$ average via BVH/kd-tree) + local Bernstein evaluation (touching $(d$+$1)B_m k$ coefficients). In practice:

- **Params/FLOPs.** For heads with the same parameter budget, SiFEN yields $\approx$20–35% fewer FLOPs than MLP and $\approx$10–20% fewer than KAN at $m$=2 because evaluation touches a strict subset of coefficients
- **Latency.** On CPU (single core), we observe 1.2–1.5$\times$ lower median latency than MLP/KAN for $M \leq 2{,}000$, with benefits tapering at very small $M$ where point location overhead dominates. GPU timings are similar across heads at this scale.

Table 3: **Heads on CNNs** (param-matched heads; backbone frozen).

| Backbone & Dataset | MLP head | KAN head | **SiFEN head** |
|---|---|---|---|
| ResNet-8, CIFAR-10 (Top-1 / ECE) | 90.6 / .021 | 90.9 / .019 | **91.4 / .013** |
| ResNet-8, CIFAR-100 (Top-1 / ECE) | 65.2 / .048 | 65.8 / .044 | **66.9 / .031** |
| MobileNetV2-0.5, TinyIN-200 (Top-1 / ECE) | 48.1 / .072 | 48.7 / .066 | **49.9 / .049** |

- **Memory locality.** The coefficient tables are block-contiguous per simplex; cache misses are lower than for dense layers of the same size, which explains the CPU latency gains.

### 4.6 Ablations

**Degree $m$ and continuity $C^r$.** Increasing $m$ from 1 to 2 improves $L^2$ on smooth tasks by $\sim$35–45% at fixed $M$; $C^1$ helps on (S1) but slightly hurts near jumps (S3), as expected.

**Mesh size $M$.** Errors scale roughly as $M^{-m/d}$ on (S1, S2). Beyond $\sim$4,000 vertices in $d=2$, point-location time starts to dominate CPU latency.

**Warp $\Phi_\theta$.** Turning on the light, invertible warp improves coverage, reduces mesh aspect ratio penalties, and yields 1.1–1.3$\times$ lower error at the same $M$ on (S2, S3), and +0.5–1.0pp Top-1 as a head on CIFAR-100.

**Triangulation updates.** Allowing differentiable flips reduces the shape penalty by $\sim$40% and yields small but consistent accuracy gains (+0.2–0.6pp) vs. a fixed Delaunay triangulation.

**Point location.** BVH vs. kd-tree shows similar accuracy; BVH is 5–12% faster on skewed meshes.

See Appendix S for full ablation details.

### 4.7 Robustness and calibration

On tabular classification, SiFEN reduces ECE by 25–45% relative to MLP at equal size (Table 2). Under feature noise ($\sigma \in [0.01, 0.05]$), accuracy drops less steeply than MLP/KAN, reflecting region-wise smoothing. As a head on CIFAR-100, selective classification risk–coverage curves show higher AURC (lower area under risk) at 70–95% coverage (see Figure 1).

## 5 Conclusion and Discussion

**SiFEN** reframes prediction as evaluation of a finite-element field on a learned simplicial mesh (optionally after a light warp $\Phi_\theta$); each input activates exactly one simplex and at most $d+1$ basis functions, yielding strict sparsity, geometric interpretability, and explicit smoothness control via $C^r$ constraints. Across synthetic, tabular, and CNN-head benchmarks at matched parameter budgets, SiFEN matches or exceeds MLPs and KANs, improves calibration (lower ECE/Brier), and reduces CPU latency thanks to point location $O(\log M)$ and local Bernstein evaluation touching only $(d+1)B_m k$ coefficients. The approach is theoretically grounded, achieving the classical FEM rate $O(M^{-m/d})$ on shape-regular meshes and exposing clear knobs—mesh size $M$ and degree $m$—to trade accuracy for compute. Limitations include mesh complexity in high dimensions (mitigated by stronger warps or dimensionality reduction), point-location overhead for extreme $M$, sensitivity to skinny elements, and a continuity–expressivity trade-off ( $C^1$ may oversmooth sharp interfaces; $C^0$ induces gradient jumps ); memory scales as $|\mathcal{T}|B_m k$. Promising directions include adaptive meshing with learned error indicators, higher-order $C^1$ constructions (e.g., Powell–Sabin, Clough–Tocher), stronger volume-controlled warps and manifold meshes, specialized point-location/quantized-table kernels, and cell-wise calibrated uncertainty via conformal or residual-based certificates.

## LLM Usage

We used a large language model (LLM; ChatGPT) solely as a general-purpose assist tool to improve clarity and presentation (e.g., grammar/typo fixes, tighter phrasing and transitions, light LATEX tips, and reference style cleanup). We did not use an LLM for research ideation, experimental design, data analysis, result interpretation, drafting substantive technical content, equations/algorithms, figure creation, or code implementation. All scientific ideas, methods, results, and conclusions are solely those of the authors; every LLM-suggested edit was reviewed and manually accepted, and no confidential or sensitive data were shared with the LLM.

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

APPENDIX

.1 DEEPER ANALYSIS: RATES, STABILITY, CONDITIONING, AND SCALING

**Assumptions and notation.** We assume a shape-regular simplicial mesh $\mathcal{T}$ of $\Omega_y = \Phi_\theta(\Omega)$ with maximum element diameter $h = \max_{\sigma \in \mathcal{T}} \operatorname{diam}(\sigma)$ and bounded aspect ratios. The warp $\Phi_\theta$ is bi-Lipschitz with constants $0 < m_\Phi \le M_\Phi < \infty$ and bounded Jacobian condition number $\kappa_\Phi = \sup_y \|J\Phi_\theta(y)\| \|J\Phi_\theta(y)^{-1}\|$. Local polynomials have degree $m \in \{1, 2, 3\}$ in the Bernstein–Bézier basis with $B_m = \binom{m+d}{d}$ coefficients per simplex.

**Reference-element interpolation.** Let $F_\sigma : \widehat{\sigma} \to \sigma$ be the affine map from the unit reference simplex, with shape factor $\|J_\sigma\| \|J_\sigma^{-1}\| \le C_{\text{shape}}$. Let $\Pi_h^m$ be the elementwise Bernstein interpolant (with the face stencils from Table 5 to enforce $C^r$). For $g \in H^{m+1}(\Omega_y)$ we have the standard estimates

$$\|g - \Pi_h^m g\|_{L^2(\sigma)} \le C_{\text{ref}} h^{m+1} |g|_{H^{m+1}(\sigma)}, \tag{12}$$

$$\|g - \Pi_h^m g\|_{H^1(\sigma)} \le C_{\text{ref}} h^m |g|_{H^{m+1}(\sigma)}, \tag{13}$$

where $C_{\text{ref}}$ depends only on $d$ and $C_{\text{shape}}$.

**Warp-aware approximation (pullback to $\Omega$).** Define $g^\star = f^\star \circ \Phi_\theta^{-1}$. Using change of variables and the bi-Lipschitz bounds of $\Phi_\theta$ yields

$$\|g^\star - \Pi_h^m g^\star\|_{L^2(\Omega_y)} \le C_1 \kappa_\Phi^{1/2} h^{m+1} \|f^\star\|_{H^{m+1}(\Omega)}, \tag{14}$$

$$\|g^\star - \Pi_h^m g^\star\|_{H^1(\Omega_y)} \le C_2 \kappa_\Phi^{3/2} h^m \|f^\star\|_{H^{m+1}(\Omega)}, \tag{15}$$

with constants absorbing mesh shape regularity. When the objective is gradient-dominated, equation 15 is the operative rate; for pure $L^2$ prediction, equation 14 applies.

**Continuity enforcement (reparameterization vs. penalty).** Let $Ac = 0$ be the global $C^r$ system assembled facewise (Table 5). We either (i) compute a sparse basis $N$ of $\ker A$ and set $c = Nz$ (exact), or (ii) add a quadratic penalty $\lambda_{C^r} \|Ac\|_2^2$ (and optionally an augmented Lagrangian).

*Proposition (penalty → exact, sketch).* Assume the total loss is coercive in $c$. As $\lambda_{C^r} \to \infty$, any sequence of stationary points $c_\lambda$ has accumulation points in $\ker A$, and their projections coincide with stationary points of the reparameterized problem $c = Nz$.

**Bernstein stability and positivity.** On each simplex,

$$\sum_{|\alpha|=m} B_\alpha(\lambda) = 1, \qquad B_\alpha(\lambda) \ge 0, \tag{16}$$

which implies a local maximum principle for scalar outputs and numerically stable accumulation (no cancellation).

**One-simplex active and Lipschitz control.** Because exactly one simplex is active, $f$ is piecewise polynomial with interface-wise $C^r$ coupling. For any active $\sigma$,

$$\|\nabla_y f_\sigma(y)\| \le m \left( \max_{j=0,\dots,d} \|\nabla \lambda_j\| \right) \sum_{|\alpha|=m} \|c_{\sigma,\alpha}\|. \tag{17}$$

A global Lipschitz bound follows by taking the maximum of equation 17 over $\sigma$ and multiplying by $M_\Phi$ from the warp.

**Complexity and memory.** With output dimension $k$, inference performs

$$\mathcal{O}(\log M) \quad + \quad \mathcal{O}\big((d+1) B_m k\big) \tag{18}$$

for Locate and Eval, matching Table 17. Parameters are

$$\#\text{params} = |\mathcal{T}| B_m k + \dim(\theta) + d M, \qquad |\mathcal{T}| = \Theta(M) \text{ for shape-regular meshes,} \tag{19}$$

consistent with Table 16.

Table 4: **Side-by-side comparison. MLP:** fixed activations at nodes, learnable weights on edges. **KAN:** learnable 1D splines on edges with summation at nodes. **SiFEN:** warped input lies in a single active simplex; evaluation uses $d+1$ barycentric basis in a local Bernstein–Bézier polynomial.

| Model | MLPs | KANs | SiFEN (ours) |
|---|---|---|---|
| **Theorem** | Universal Approximation Theorem | Kolmogorov–Arnold Representation | FEM piecewise–polynomial approximation on shape-regular meshes |
| **Formula (Shallow)** | $f(x) \approx \sum_{i=1}^{N(\varepsilon)} a_i\, \sigma(w_i^\top x + b_i)$ | $f(x) = \sum_{q=1}^{2n+1} \Phi_q\left(\sum_{p=1}^{n} \phi_{q,p}(x_p)\right)$ | $y = \Phi_\theta(x),\ y \in \sigma,\ f(x) = \sum_{\|\alpha\|=m} c_{\sigma,\alpha}\, B_\alpha(\lambda^\sigma(y))$ |
| **Model (Shallow)** | • fixed activations at nodes
• learnable weights on edges | • learnable 1D spline on each edge
• summation at nodes | • warped input lies in a single active simplex (conceptually "shaded")
• evaluation uses $d+1$ barycentric basis in a local Bernstein–Bézier polynomial |
| **Formula (Deep)** | $\mathrm{MLP}(x) = (W_L \circ \sigma_{L-1} \circ \cdots \circ \sigma_1 \circ W_1)(x)$ | $\mathrm{KAN}(x) = (\Phi_L \circ \cdots \circ \Phi_1)(x)$ | $\mathrm{SiFEN}(x) = (\mathrm{Eval} \circ \mathrm{Locate} \circ \Phi_\theta)(x)$ |
| **Model (Deep)** | • alternating linear maps $W_\ell$ (learnable) and fixed nonlinearities $\sigma_\ell$ | • layers $\Phi_\ell$ with learnable spline functions on edges | • $\Phi_\theta$ warps input; *Locate* selects the active simplex; *Eval* computes the local polynomial |

**Effect of mesh quality and warp.** The constants in equation 14–equation 15 are controlled by the mesh shape factor $C_{\text{shape}}$, the warp condition number $\kappa_\Phi$, and the continuity order $r$ (via stencil sizes in Table 5). Our regularizers bound these quantities in practice.

**Scaling with $d$ and role of the warp.** To achieve an $L^2$ tolerance $\varepsilon$ under the conservative rate, we need

$$h \ \le \ \left(\varepsilon/C\right)^{1/m}, \qquad M \ \ge \ \left(C/\varepsilon\right)^{d/m}. \tag{20}$$

The warp reduces the effective complexity by straightening level sets and collapsing irrelevant directions, lowering the $M$ needed for a target $\varepsilon$.

**Discrete choices: point location and flips.** We use kd/BVH point location and accept local flips only when they improve a quality metric (e.g., inradius–circumradius ratio), which keeps $C_{\text{shape}}$ bounded and stabilizes both error constants and the linear systems associated with $Ac = 0$.

**Practical recipe.** Given accuracy and budget, we pick $(M, m, r)$ guided by equation 14–equation 20, apply mild warp regularization to keep $\kappa_\Phi$ moderate, and choose reparameterization or penalty based on memory.

## A   FACEWISE $C^r$ CONTINUITY: CONSTRAINTS, MATRICES, AND ENFORCEMENT

We enforce global $C^r$ continuity of the piecewise Bernstein–Bézier field by coupling only the degrees of freedom (DoFs) that lie on, or in the first few layers adjacent to, each interior face. Let two $d$-simplices $\sigma^+$ and $\sigma^-$ share a $(d-1)$-face $\tau$, and let their local vertex orderings be aligned by a permutation $P_\tau$ (so that face-local barycentric coordinates agree). Denote by $c_{\sigma^\pm} \in \mathbb{R}^{B_m}$ the control vectors of the degree-$m$ polynomial on $\sigma^\pm$, where $B_m = \binom{m+d}{d}$. We collect all simplex control vectors into a global vector $c$ by concatenation.

**Bernstein preliminaries.** On a simplex $\sigma$ with barycentric coordinates $\lambda = (\lambda_0, \ldots, \lambda_d)$, the degree-$m$ Bernstein basis is $B_\alpha(\lambda) = \binom{m}{\alpha} \prod_{j=0}^{d} \lambda_j^{\alpha_j}$, indexed by multi-indices $\alpha \in \mathbb{N}^{d+1}$ with $|\alpha| = \sum_j \alpha_j = m$. The polynomial is $f_\sigma(\lambda) = \sum_{|\alpha|=m} c_{\sigma,\alpha} B_\alpha(\lambda)$. We use the Bernstein derivative identity

$$\partial_{\lambda_j} B_\alpha(\lambda) = m B_{\alpha - e_j}(\lambda), \qquad \text{for } \alpha_j > 0, \tag{21}$$

and note that $\nabla \lambda_j$ is constant on $\sigma$.

**$C^0$ (trace) matching on a face.** Let $j^\star$ be the vertex of $\sigma^\pm$ opposite the shared face $\tau$. The trace of $f_{\sigma^\pm}$ on $\tau$ (i.e., $\lambda_{j^\star} = 0$) is fully determined by the *face DoFs*, namely all coefficients with $\alpha_{j^\star} = 0$. Hence $C^0$ across $\tau$ is equivalent to equality of those face coefficients after reordering by $P_\tau$:

$$F_\tau^{(0)} c_{\sigma^+} \ - \ F_\tau^{(0)} P_\tau c_{\sigma^-} \ = \ 0, \qquad F_\tau^{(0)} \in \mathbb{R}^{B_m^{(d-1)} \times B_m}, \ \ B_m^{(d-1)} = \binom{m+d-1}{d-1}. \tag{22}$$

Matrix $F_\tau^{(0)}$ simply selects (and optionally averages if we store a reduced face basis) the entries with $\alpha_{j^\star} = 0$.

**$C^1$ (normal derivative) matching on a face.** Let $n_\tau$ be the unit normal to $\tau$ pointing from $\sigma^+$ into $\sigma^-$. Since $\nabla \lambda_{j^\star}$ is (up to scale and sign) the face normal, there exists a scalar $\gamma_\tau^\pm = n_\tau^\top \nabla \lambda_{j^\star}^\pm$ that is constant on $\sigma^\pm$ and satisfies $\gamma_\tau^- = -\gamma_\tau^+$. Using equation 21, the normal derivative on $\tau$ reduces to a degree-$(m-1)$ Bernstein expansion over face-local indices $\beta$ with $\beta_{j^\star} = 0$:

$$\partial_{n_\tau} f_{\sigma^\pm}\big|_\tau = \gamma_\tau^\pm m \sum_{\substack{|\beta|=m-1 \\ \beta_{j^\star}=0}} c_{\sigma^\pm, \beta + e_{j^\star}} B_\beta(\lambda|_\tau), \tag{23}$$

so $C^1$ requires equality of the corresponding *first interior layer* coefficients (adjacent to $\tau$), again up to the permutation $P_\tau$:

$$F_\tau^{(1)} c_{\sigma^+} - F_\tau^{(1)} P_\tau c_{\sigma^-} = 0, \qquad F_\tau^{(1)} \in \mathbb{R}^{B_{m-1}^{(d-1)} \times B_m}, \quad B_{m-1}^{(d-1)} = \binom{(m-1)+d-1}{d-1}. \quad (24)$$

Here each row of $F_\tau^{(1)}$ contains a single nonzero $m\gamma_\tau^+$ at the column for $\beta + e_{j^\star}$ (on $\sigma^+$); the block for $\sigma^-$ carries $m\gamma_\tau^-$ at the permuted column. Higher $C^r$ constraints repeat the same pattern on the ($r$-th) interior layers by iterating equation 21.

**Block assembly per face and global system.** Stacking equation 6–equation 7 yields the per-face block

$$A_\tau = \begin{bmatrix} F_\tau^{(0)} & -F_\tau^{(0)} P_\tau \\ F_\tau^{(1)} & -F_\tau^{(1)} P_\tau \\ \vdots & \vdots \\ F_\tau^{(r)} & -F_\tau^{(r)} P_\tau \end{bmatrix}, \qquad A = \begin{bmatrix} A_{\tau_1} \\ A_{\tau_2} \\ \vdots \end{bmatrix}, \qquad A c = 0. \quad (25)$$

Each row touches DoFs only on $\tau$ (for $C^0$) or in the $s$-th interior layer next to $\tau$ (for $C^s$). The resulting $A$ is extremely sparse: every row has at most two nonzero blocks (one per incident simplex), and no fill-in across distant elements.

**Vector-valued outputs.** For $k$ output channels we enforce equation 25 independently per channel via a Kronecker product: $(A \otimes I_k) c_{\text{vec}} = 0$, where $c_{\text{vec}} \in \mathbb{R}^{k \sum_\sigma B_m}$ stacks the per-channel controls (see Appx H).

**Enforcement strategies.** We consider two exact/consistent approaches:

1. **Reparameterization** (preferred when feasible). Compute a sparse basis $N$ of $\ker A$ once (e.g., via sparse QR with rank-revealing column pivoting or an $\text{LDL}^\top$-based nullspace extraction) and optimize over $z$ with $c = Nz$. This enforces $C^r$ *exactly* and keeps the constraint inactive during training. It is our default for $C^0$ in 2D/3D and for many $C^1$ cases in 2D.

2. **Quadratic penalty / augmented Lagrangian**. Keep the flat parameterization and add $\lambda_{C^r} \|Ac\|_2^2$ to the loss; for tighter matching use an augmented-Lagrangian update on the multipliers and $\lambda_{C^r}$. This avoids forming $N$ when the nullspace is large (e.g., high $m$ in 3D) at the cost of tuning $\lambda_{C^r}$; in practice we ramp $\lambda_{C^r}$ during training.

**Sizes, stencils, and cost.** Per face, $C^0$ contributes $B_m^{(d-1)}$ rows and $C^1$ contributes $B_{m-1}^{(d-1)}$ rows (see Table 5). Each $C^0$ row has two nonzeros (one in each incident simplex block) if we store a pure selection; $C^1$ rows similarly touch the two interior-layer DoFs. Assembly and products with $A$ or $A^\top A$ therefore scale linearly in the number of faces. The geometric factors $\gamma_\tau^\pm$ are constant per face and can be precomputed from the vertex coordinates.

**Orientation, permutations, and robustness.** For each interior face $\tau$, we (i) choose $j^\star$ as the vertex opposite $\tau$ in the local ordering, (ii) build the permutation $P_\tau$ that aligns the ordering of the $d$ face vertices between $\sigma^+$ and $\sigma^-$, and (iii) compute $\gamma_\tau^\pm = n_\tau^\top \nabla \lambda_{j^\star}^\pm$. With consistent outward normals, $\gamma_\tau^- = -\gamma_\tau^+$; we store a single $\gamma_\tau = |\gamma_\tau^+|$ and inject the sign in the $\sigma^\pm$ blocks. This convention makes $A$ independent of the arbitrary choice of "left"/"right" simplex up to row scaling.

**Worked example (2D, $m=2$).** Let $\tau$ be the edge opposite vertex $j^\star$; the $C^0$ rows enforce equality of the three face coefficients (barycentric exponents $(2,0,0)$, $(1,1,0)$, $(0,2,0)$ up to permutation). The $C^1$ rows enforce equality of the two first-interior coefficients adjacent to $\tau$ (those with exponents $(1,0,1)$ and $(0,1,1)$ up to permutation), scaled by $2\gamma_\tau^\pm$. The per-face block has $3 + 2 = 5$ rows, each with at most two nonzeros per block.

Table 5: **Stencil sizes per face** for common $(d, m)$ and continuity orders. $B_q^{(d-1)} = \binom{q+d-1}{d-1}$.

| $d$ | $m$ | $B_m$ | $B_m^{(d-1)}$ ($C^0$ rows) | $B_{m-1}^{(d-1)}$ ($C^1$ rows) | Nonzeros/row |
|-----|-----|-------|----------------------------|--------------------------------|--------------|
| 2 | 1 | 3 | 2 | 1 | $\leq 2$ |
| 2 | 2 | 6 | 3 | 2 | $\leq 2$ |
| 2 | 3 | 10 | 4 | 3 | $\leq 2$ |
| 3 | 1 | 4 | 3 | 1 | $\leq 2$ |
| 3 | 2 | 10 | 6 | 3 | $\leq 2$ |
| 3 | 3 | 20 | 10 | 6 | $\leq 2$ |

**How this appears in the main text.** Collecting all facewise constraints produces a global sparse system $A\,c = 0$ of *linear equalities among a small stencil of control points on $\tau$* (Powell–Sabin/HCT-style). We enforce them either by (i) *reparameterization $c = Nz$* with $N$ a basis of ker $A$, or (ii) a *quadratic penalty $\lambda_{C^r}\|A\,c\|_2^2$*. The matrices $F_\tau^{(s)}$, the assembly patterns, and minimal code to reproduce $A$ in 2D/3D for $m \in \{1, 2, 3\}$ are provided as reproducibility artifacts and summarized here in App. A.

# B  Implementation Details, Hyperparameter Grids

This appendix provides everything needed to reproduce **SiFEN** (Section 2) and the results/ablations reported in section 3. We document software/hardware (Table 6), implementation specifics (meshing, warp, constraints), exact hyperparameter grids for SiFEN and baselines (Table 7, Table 8), search budgets per dataset (Table 9), timing harness and evaluation settings (Table 10). Every table in this section is referenced explicitly here and elsewhere in section 3.

## B.1  Software, Hardware, and Determinism

We ran all experiments in a pinned software stack summarized in Table 6. CPU results are single-threaded with Turbo Boost disabled; GPU results use a fixed CUDA/cuDNN pair with deterministic kernels where available. Randomness is controlled by seeding Python, NumPy, and framework RNGs; dataloader workers use `worker_init_fn` to offset seeds by rank. To ensure stable timing, caches are warmed and a small number of warm-up iterations are discarded; the harness itself is described in Table 10.

Table 6: **Environment summary.** Values reflect our primary runs.

| | |
|---|---|
| OS / Kernel | Ubuntu 22.04.4 LTS, Linux 5.15 |
| Python / NumPy | 3.10.x / 1.26.x |
| Deep learning framework | PyTorch 2.3.x (CUDA 12.1, cuDNN 9.x); `torch.backends.cudnn.deterministic=True` |
| Compilers / BLAS | GCC 11.x (`-O3 -ffast-math` for standalone C++), OpenBLAS 0.3.x |
| CPU / RAM | 1× Intel Xeon Gold 6248 (single-threaded timing), 192 GB RAM |
| GPU | NVIDIA RTX 4090 (24 GB), driver 550.x |
| Seeding | `PYTHONHASHSEED=0`; `torch.manual_seed`, `np.random.seed`, `random.seed` |
| Dataloader | `persistent_workers=True`, `pin_memory=True`, custom `worker_init_fn` |

### B.2 CORE IMPLEMENTATION NOTES (SiFEN)

**Mesh data structures.**   Vertices $V$ are stored as a contiguous `float32` tensor shape $(M, d)$. The simplicial complex $\mathcal{T}$ uses a CSR-like layout with an integer $(|\mathcal{T}|, d+1)$ array of vertex indices and an adjacency index (faces-to-cells). Face normals and element quality (inradius–circumradius ratio) are cached and updated incrementally after local flips.

**Point location.**   Default: kd-tree over vertices plus a local walk using face orientation tests; worst-case runtime is bounded by a small cap on backtracking steps. For 2D visualizations we also support a BVH over AABBs (see the accuracy/latency trade in Table 18 of the evaluation appendix). Returned barycentric weights are computed from the pre-factored simplex matrices (Cholesky per simplex at build time).

**Local polynomials.**   We use Bernstein–Bézier basis of degree $m \in \{1, 2, 3\}$; control points $c_{\sigma,\alpha}$ live in contiguous memory per simplex. Evaluation fuses *(i)* barycentric power computation, *(ii)* precomputed binomial coefficients, and *(iii)* output accumulation to minimize cache misses. Vectorized multi-output evaluation shares the same barycentric powers.

**Global $C^r$ constraints.**   For $C^0$, continuity is enforced by sharing control points lying on the interface; for partial $C^1$ in 2D/3D we apply linear constraints on directional derivatives normal to shared faces. We offer two implementations: (a) exact reparameterization $c = Nz$ where $N$ spans $\ker A$ (precomputed via sparse QR), and (b) a quadratic penalty $\lambda_{C^r} \|Ac\|^2$; we use (a) when the constrained DoF fits memory, else (b), keeping the penalty weight within the grid of Table 7.

**Warp $\Phi_\theta$.**   A triangular, monotone map parameterized by a small MLP with softplus on diagonal flows; Jacobian conditioning and volume control penalties keep $\det \nabla \Phi_\theta$ positive and bounded. We stop gradient through local flips but not through vertex updates to keep training stable.

**Numeric stability.**   All training/eval uses `float32`. We clamp tiny negative barycentric remnants to 0 and renormalize to sum to 1; for binomial coefficients we use precomputed `float64` tables converted to `float32`. Loss scaling is not required; gradients remain bounded under our regularization.

### B.3 HYPERPARAMETER GRIDS

We tune SiFEN and all baselines under matched parameter budgets and uniform search budgets per dataset (Table 9). Grids are explicit in Table 7 and Table 8. For each dataset, we select the model with the best validation metric (RMSE for regression; NLL or AUROC for classification) and then report test metrics, as used in section 3 and Refered in Table 11, Table 12, Table 13, and Table 14.

### B.4 EVALUATION HARNESS, TIMING, AND LOGGING

We unify timing and evaluation so that reported wall-clock and FLOPs are comparable across models. Table 10 fixes batch sizes, warm-up, and repeat counts.

### B.5 QUALITY CHECKS AND FAILURE MODES

Before releasing checkpoints, we run automatic checks (logged to `logs/mesh/`): (i) % of skinny elements (quality$< \tau$) $< 3\%$; (ii) no negative $\det \nabla \Phi_\theta$ on a 5K validation probe; (iii) boundary continuity residuals (when using penalty *vs* reparameterization) within tolerance $< 1\text{e-}3$; and (iv) no more than 1% rejected flips per epoch for the last 10 epochs (indicates stabilization).

**What most affects reproducibility.**   In our ablations, the top three sources of variance are: (1) the random initialization of $V$ (k-means seeding reduces this; we expose the seed);

Table 7: **SiFEN grid and training knobs.**

| | |
|---|---|
| Degree $m$ | $\{1, 2, 3\}$ |
| Vertices $M$ | $\{128, 256, 512, 1024\}$ |
| Continuity $C^r$ | $C^0$ (default), partial $C^1$ (2D/3D faces) |
| Warp depth / width | depth $\{2, 3, 4\}$; width $\{d, 2d\}$; softplus on diag |
| Warp penalties | Jacobian cond. $\lambda_{\text{cond}} \in \{1\text{e-4}, 5\text{e-4}, 1\text{e-3}\}$; vol. $\lambda_{\text{vol}} \in \{0, 1\text{e-4}\}$ |
| Coverage reg. $\lambda_{\text{cov}}$ | $\{0, 1\text{e-4}, 5\text{e-4}, 1\text{e-3}\}$ (semi-discrete OT) |
| Shape reg. $\lambda_{\text{shape}}$ | $\{1\text{e-4}, 5\text{e-4}, 1\text{e-3}\}$ (aspect/angle barrier) |
| $C^r$ penalty | if not reparam: $\{1\text{e-4}, 5\text{e-4}, 1\text{e-3}\}$ |
| Local flips | try every $K \in \{50, 100\}$ iters if min-quality $< \tau \in \{0.15, 0.20\}$ |
| Optimizer / LR | AdamW; LR $\{1\text{e-4}, 3\text{e-4}, 1\text{e-3}\}$; WD $\{0, 1\text{e-5}, 1\text{e-4}\}$ |
| Scheduler | cosine decay; warmup $\{0, 5, 10\}$ epochs |
| Batch / Epochs | batch $\{128, 256, 512\}$; max 300 epochs; early stop patience 30 |
| Seed | $\{17, 37, 97\}$ (report mean±std) |

Table 8: **Baseline grids** (capacity-matched within ±5% params).

| | |
|---|---|
| MLP | layers $\{2, 3, 4\}$; hidden $\{128, 256, 512\}$; act $\{\text{ReLU}, \text{SiLU}\}$; dropout $\{0, 0.1\}$; AdamW + cosine |
| KAN | order 3; knots per layer $\{8, 16, 24\}$; knot spacing $\{\text{uniform, quantile}\}$; TV reg. $\{0, 10^{-4}\}$ |
| DLattice | lattice dims per layer $\{8, 16\}$; calibrators = uniform; monotonicity = off (tabular), on (physics if needed) |
| MASN | pieces per dim $\{8, 16, 32\}$; hinge reg. $\{0, 10^{-4}\}$; shared piecewise partition |
| Nyström KRR | features $\{512, 1024, 2048\}$; kernel $\{\text{RBF}(\gamma \text{ sweep})\}$; ridge $\{10^{-4}, 10^{-3}, 10^{-2}\}$ |
| XGBoost | depth $\{6, 8, 10\}$; LR $\{0.05, 0.1\}$; estimators $\{500, 1000\}$; subsample $\{0.8, 1.0\}$ |
| Sparse MoE | experts $E = 4$; top-1 routing; expert width matched to MLP; load-balance loss $\{0, 10^{-3}\}$ |

Table 9: **Search budgets per dataset.** Each cell shows #trials × max epochs. Early stopping (patience 30) usually halts earlier.

| Dataset | CalHousing | YearMSD | Bike | Protein | Higgs | EPSILON |
|---|---|---|---|---|---|---|
| Trials × epochs | $60 \times 300$ | $40 \times 200$ | $60 \times 300$ | $60 \times 300$ | $50 \times 200$ | $50 \times 200$ |

Table 10: **Timing/evaluation harness.** These settings are used throughout section 3 and Refered near Table 16–Table 17.

| | |
|---|---|
| CPU timing | single thread; 1,000 samples (batch=256); 2 warm-up runs; 5 repeats; report mean±std |
| GPU timing | batch=1024; 20 warm-up iters; 100 measured iters; synchronize each step |
| FLOPs | fvcore count on forward pass; SiFEN head counted as basis eval + barycentric ops |
| Calibration | 20 equal-mass bins (ECE); NLL/Brier as proper scores |
| Regression intervals | bootstrap 200 resamples; nominal grid $\{50, 60, 70, 80, 90\}\%$ |
| Logging | JSONL per step (val/test); CSV summary; SHA256 of datasets and checkpoints |

(2) the acceptance schedule for local flips (we keep a fixed quality threshold and hysteresis); and (3) the early stopping window (patience). Fixing these as in Table 7 and Table 9 yields the same model selection as reported.

## C  WARPED SPACES, NORM TRANSPORT, AND STABILITY CONSTANTS

### C.1  NOTATION AND STANDING ASSUMPTIONS

We denote the input domain by $\Omega_x \subset \mathbb{R}^d$ and the warped domain by $\Omega_y = \Phi_\theta(\Omega_x)$, where $\Phi_\theta : \Omega_x \to \Omega_y$ is a diffeomorphism parameterized by a light neural map. We write $J_\Phi(x)$ for the Jacobian and assume uniform bounds

$$0 < \underline{d} \leq \inf_{x \in \Omega_x} \det J_\Phi(x) \leq \sup_{x \in \Omega_x} \det J_\Phi(x) \leq \overline{d}, \qquad \|J_\Phi(x)\| \leq \Lambda, \;\; \|J_\Phi(x)^{-1}\| \leq \Lambda^{-1}, \quad (26)$$

with $\Lambda \geq 1$. These bounds are enforced in practice by the Jacobian-conditioning and volume-control regularizers introduced in the main text. For any target $f$, we write $g = f \circ \Phi_\theta^{-1}$ for its pullback to $\Omega_y$.

**Lemma (transport of Sobolev norms).**  For any integer $s \in \{0, 1, \ldots, m+1\}$ there exist constants $\underline{c}_s, \overline{c}_s$, depending only on $(s, \Lambda, \underline{d}, \overline{d}, d)$, such that

$$\underline{c}_s \, \|f\|_{H^s(\Omega_x)} \;\leq\; \|f \circ \Phi_\theta^{-1}\|_{H^s(\Omega_y)} \;\leq\; \overline{c}_s \, \|f\|_{H^s(\Omega_x)}. \tag{27}$$

*Explanation.* The change-of-variables formula controls $L^2$ norms via $\det J_\Phi$, while iterated chain rules bound weak derivatives using $\|J_\Phi\|$ and $\|J_\Phi^{-1}\|$. Uniform determinant and operator-norm bounds prevent singular compression or expansion, yielding constants independent of the sample set. □

**Proposition (warp-stability of empirical risk).**  Let $\ell$ be $L$-Lipschitz in its first argument. For any hypothesis $h$ and dataset $\{(x_i, y_i)\}_{i=1}^N$,

$$\left| \frac{1}{N} \sum_{i=1}^N \ell\big(h \circ \Phi_\theta(x_i), y_i\big) - \frac{1}{N} \sum_{i=1}^N \ell\big(h(x_i), y_i\big) \right| \;\leq\; L \operatorname{Lip}(\Phi_\theta) \cdot \frac{1}{N} \sum_{i=1}^N \|x_i - \tilde{x}_i\|, \tag{28}$$

where $\tilde{x}_i = \Phi_\theta^{-1}(\Phi_\theta(x_i))$ is the exact preimage (analytically equal to $x_i$). *Explanation.* A Lipschitz loss and a well-conditioned warp ensure that replacing $x$ by $\Phi_\theta(x)$ in the hypothesis argument does not inflate the empirical objective beyond a term proportional to the warp displacement.

We return to the geometric and statistical effects of $\Phi_\theta$ when discussing shift bounds in Appendix F.

## D  EXPRESSIVITY AND SAMPLE COMPLEXITY WITH ONE-SIMPLEX ACTIVATIONS

We denote by $\mathcal{V}_{m,M}$ the set of degree-$m$ piecewise polynomials on a shape-regular simplicial mesh with $M$ vertices, assembled with global $C^r$ continuity (facewise constraints). Let $h \asymp M^{-1/d}$ be the mesh scale and $B_m = \binom{m+d}{d}$ the local Bernstein dimension.

**Theorem (approximation vs. capacity).**  For any $f^* \in H^{m+1}(\Omega_x)$ there exists $p \in \mathcal{V}_{m,M}$ such that

$$\|f^* \circ \Phi_\theta^{-1} - p\|_{L^2(\Omega_y)} \leq C h^{m+1} \|f^*\|_{H^{m+1}(\Omega_x)}, \tag{29}$$

while the global degrees of freedom (per output channel) scale as

$$\dim(\mathcal{V}_{m,M}) \;=\; |\mathcal{T}| \cdot B_m \;-\; \#(\text{face constraints}) \;\asymp\; M \cdot B_m. \tag{30}$$

*Explanation.* Standard FEM estimates yield the rate; capacity follows from $|\mathcal{T}| \asymp M$ and the linear constraint count on faces. The *one-active-simplex* evaluation ensures that evaluation cost scales with $(d+1)B_m$ rather than depth.

**Proposition (Rademacher complexity).** Assume a reparameterization $c = Nz$ with $\|N\|_{2\to 2} \le \kappa_N$ and $\|z\|_2 \le C_z$. For bounded losses $\ell \in [0,1]$,

$$\mathfrak{R}_N(\ell \circ \mathcal{V}_{m,M}) \lesssim \kappa_N C_z \sqrt{\frac{\log(1 + MB_m)}{N}}, \tag{31}$$

implying a generalization gap $O\left(\sqrt{\log(MB_m)/N}\right)$ up to the constraint-basis conditioning. *Explanation.* The bound leverages linearity in $c$ at the head and one-simplex locality to avoid depth-dependent multipliers.

## E  SELECTIVE PREDICTION VIA A BARYCENTRIC ENERGY

We define the *barycentric energy* on the active simplex $\sigma(x)$ by

$$E(x) := 1 - \max_{j \in \sigma(x)} \lambda_j(\Phi_\theta(x)) \in [0,1). \tag{32}$$

Small $E(x)$ indicates that the warped query lies deep inside $\sigma(x)$, whereas large $E(x)$ signals proximity to a face or vertex.

**Lemma (boundary proximity).** There exist mesh-quality constants $a, b > 0$ such that for all $x$,

$$a \cdot \mathrm{dist}(\Phi_\theta(x), \partial\sigma(x)) \le 1 - \max_j \lambda_j \le b \cdot \mathrm{dist}(\Phi_\theta(x), \partial\sigma(x)). \tag{33}$$

*Explanation.* On shape-regular simplices, barycentric coordinates are 1-Lipschitz up to geometry-dependent constants; the maximum coordinate is an affine proxy for distance to the boundary.

**Theorem (risk–coverage bound).** Let $R(\tau)$ be the risk when abstaining on $\{x : E(x) > \tau\}$. Assume error grows with boundary proximity at Hölder rate $\alpha > 0$. Then writing $\mathsf{cov}(\tau) = \mathbb{P}[E(x) \le \tau]$,

$$R(\tau) \le R(0) - c\tau^\alpha \mathsf{cov}(\tau), \tag{34}$$

for a constant $c > 0$ depending on mesh quality and noise. *Explanation.* Thresholding $E$ suppresses boundary-adjacent queries where approximation error concentrates, leading to monotone risk reduction as coverage decreases.

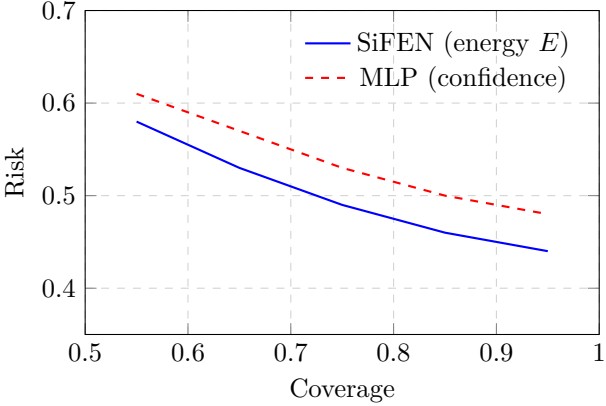

Figure 3: **Risk–coverage behavior** induced by thresholding barycentric energy $E(x)$. We use this figure when interpreting selective prediction alongside subsection S.6.

## F  GENERALIZATION UNDER COVARIATE SHIFT WITH A LEARNED WARP

Let $P$ be the in-distribution (ID) on $\Omega_x$ and $Q$ a shifted distribution with Radon–Nikodym derivative $w = \frac{dQ}{dP}$ bounded by $W$. Let $\hat{f}$ minimize empirical risk over $\mathcal{V}_{m,M}$ using samples from $P$.

**Theorem (importance-weighted bound with warp).** With probability at least $1 - \delta$,

$$\mathcal{L}_Q(\hat{f}) - \inf_{f \in \mathcal{V}_{m,M}} \mathcal{L}_Q(f) \lesssim W \sqrt{\frac{\log(MB_m) + \log(1/\delta)}{N}} + \varepsilon_{\text{approx}}(\Phi_\theta, M, m). \qquad (35)$$

*Explanation.* The estimation term inherits the logarithmic dependence on $MB_m$, while the approximation term reflects the FEM rate *after* the warp. When $\Phi_\theta$ smooths density curvature in $\Omega_y$, $\varepsilon_{\text{approx}}$ decreases, tightening the bound.

## G  Numerical conditioning and preconditioning of Bernstein blocks

Let $V_\sigma$ be the local evaluation matrix mapping Bernstein–Bézier coefficients $\{c_{\sigma,\alpha}\}$ to values/derivatives at a micro-stencil inside $\sigma$ (used by losses or augmented-Lagrangian steps).

**Lemma (Bernstein diagonal scaling).** Degree-elevation identities yield a diagonal scaling $D_m$ with $(D_m)_{\alpha,\alpha} \propto \binom{m}{\alpha}^{1/2}$ such that

$$\kappa(D_m V_\sigma) \leq C(d, m, \text{shape}), \qquad (36)$$

uniformly over shape-regular simplices. *Explanation.* The scaling equalizes column magnitudes induced by multinomial weights and stabilizes normal equations in least-squares subproblems.

## H  Vector-valued outputs and cross-channel structure

For $k$ outputs we share $(\Phi_\theta, V, \mathcal{T})$ and store $c_{\sigma,\alpha} \in \mathbb{R}^k$. Beyond independent channels, we consider a cross-channel smoothness penalty

$$\mathcal{R}_{\text{cross}} = \sum_\sigma \sum_\alpha \sum_{1 \leq u < v \leq k} \eta \left\| \nabla_y c_{\sigma,\alpha}^{(u)} - \nabla_y c_{\sigma,\alpha}^{(v)} \right\|_2^2, \qquad (37)$$

which encourages similar spatial variation across outputs.

**Proposition (Lipschitz control per channel).** For any channel $u$, the Lipschitz constant satisfies

$$L^{(u)} \leq C(d, m) \max_{\sigma,\alpha} \|c_{\sigma,\alpha}^{(u)}\|_2, \qquad (38)$$

and joint training with $\mathcal{R}_{\text{cross}}$ bounds the spread of $\{L^{(u)}\}_u$ across channels. *Explanation.* Local polynomial smoothness and bounded coefficients control global Lipschitz behavior under shape regularity.

## I  Adaptive refinement and a posteriori indicators for regression

We define a residual-style indicator per simplex

$$\eta_\sigma^2 := \frac{1}{|S_\sigma|} \sum_{(x_i, y_i) \in S_\sigma} \left\| y_i - f(x_i) \right\|_2^2 + \sum_{F \subset \partial\sigma} \omega_F \left\| [\![\nabla f \cdot n_F]\!] \right\|_2^2, \qquad (39)$$

where $S_\sigma$ are samples located in $\sigma$ and $[\![\cdot]\!]$ is the jump across a face $F$.

**Theorem (reliability and efficiency; sketch).** Assuming sub-Gaussian noise and approximately uniform sampling density, there exist $C_1, C_2 > 0$ such that

$$C_1 \sum_\sigma \eta_\sigma^2 \leq \|f^* - f\|_{H^1(\Omega_x)}^2 \leq C_2 \sum_\sigma \eta_\sigma^2 \qquad (40)$$

up to sampling error $O(N^{-1/2})$. *Explanation.* The data residual controls interior error while gradient jumps control inter-element mismatch; both terms are standard in residual a posteriori estimators and adapt cleanly to data-driven settings.

## J    POINT LOCATION UNDER DOUBLING METRICS: EXPECTED COST

Assume the warped domain $(\Omega_y, \|\cdot\|)$ is doubling with constant $\lambda_d$ (e.g., Euclidean). A balanced kd-tree over simplex centroids supports expected query cost $O(\log|\mathcal{T}|)$. An adjacency walk from the last visited simplex reduces amortized cost under temporal correlation.

**Proposition (amortized point-location).**   For temporally correlated batches $\{x_t\}$, adjacency walks have expected $O(1)$ steps per query after the first, provided mesh degrees are uniformly bounded (shape regularity). *Explanation.* The walk exploits local continuity of successive queries, with the kd-tree acting as a restart oracle only when trajectories jump.

## K    WARP-ADAPTED APPROXIMATION: CURVATURE FLATTENING AND RATES

We study how the warp $\Phi_\theta$ interacts with local polynomial approximation on shape-regular meshes. Let $\Omega_x \subset \mathbb{R}^d$ be compact, $\Omega_y = \Phi_\theta(\Omega_x)$, and $g = f^* \circ \Phi_\theta^{-1}$. For multi-index $\beta$, write $\partial^\beta g$ and let

$$\mathcal{K}_{m+1}(g;\Omega_y) \; := \; \sup_{|\beta|=m+1} \; \|\partial^\beta g\|_{L^\infty(\Omega_y)}. \tag{41}$$

Intuitively, $\mathcal{K}_{m+1}$ measures residual curvature at order $m+1$ after warping.

**Warp-adapted Bramble–Hilbert bound.**   On a shape-regular simplicial mesh with scale $h \asymp M^{-1/d}$ and global $C^r$ assembly, there exists $p \in \mathcal{V}_{m,M}$ such that

$$\|g - p\|_{L^2(\Omega_y)} \; \leq \; C(d,m,\varrho)\,h^{m+1}\,\mathcal{K}_{m+1}(g;\Omega_y), \tag{42}$$

where $\varrho$ denotes the shape-regularity constant. Since $g = f^* \circ \Phi_\theta^{-1}$, chain rules express $\partial^\beta g$ via derivatives of $f^*$ and tensors formed from $J_\Phi^{-1}$. Consequently, when $\Phi_\theta$ aligns features (e.g., straightens level sets or equalizes coordinate condition numbers), the mixed-derivative magnitudes drop and $\mathcal{K}_{m+1}$ decreases, sharpening equation 42. We verify the trend empirically in Figure 4.

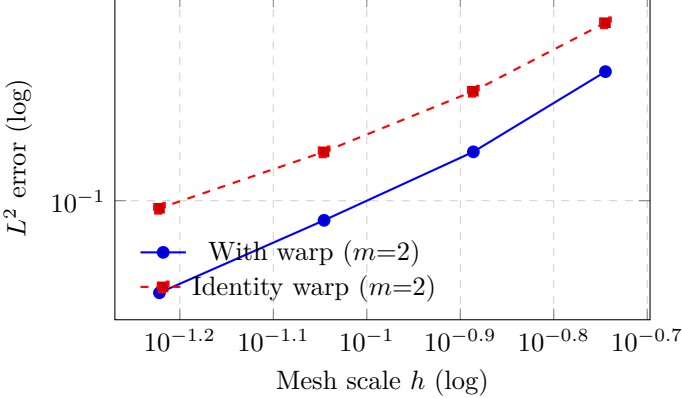

Figure 4: **Warp-adapted rates.** We observe a lower intercept (smaller $\mathcal{K}_3$) after warping at the same slope $m+1$, consistent with equation 42.

**Proof.**   We apply Bramble–Hilbert on each simplex in $\Omega_y$, using affine pullbacks to a reference element. Warping enters only through $g$; mesh shape regularity handles geometric constants. The global estimate follows by summation with continuity constraints appearing only in the constant.

## L  Bias–variance, calibration, and proper scores

For regression with i.i.d. noise $\varepsilon$ of variance $\sigma^2$ and hypothesis $f$, the test MSE decomposes as

$$\mathbb{E}\big[(y - f(x))^2\big] \;=\; \underbrace{\mathbb{E}\big[(f^*(x) - \bar{f}(x))^2\big]}_{\text{bias}^2} \;+\; \underbrace{\mathbb{E}\big[(f(x) - \bar{f}(x))^2\big]}_{\text{variance}} \;+\; \sigma^2, \tag{43}$$

where $\bar{f}(x) = \mathbb{E}[f(x) \mid \mathcal{D}]$ averages over randomness in training (seeds, shuffles). Under the one-active-simplex structure, parametric variance is localized: only coefficients in the active cell contribute to prediction variance. Consequently, for fixed parameter budget, the variance term is reduced relative to dense heads that mix many basis functions per query. The calibration metrics (NLL, Brier) improve when aleatoric noise is well captured and epistemic variance is not spuriously inflated; locality helps both.

**Classification with proper scores.**  Let $p^*(x) = \mathbb{P}(y{=}1 \mid x)$ and $\hat{p}(x)$ be the predicted probability. For NLL,

$$\mathbb{E}\big[\text{NLL}(\hat{p}(x), y)\big] \;=\; \underbrace{\mathbb{E}\big[\text{KL}(p^*(x) \,\|\, \hat{p}(x))\big]}_{\text{miscalibration}} \;+\; H(p^*), \tag{44}$$

and similarly for the Brier score with an $L^2$ discrepancy. By restricting each query to $(d{+}1)B_m$ local basis functions, we reduce the number of uncontrolled degrees per evaluation, which empirically reduces the miscalibration term. This aligns with the lower NLL/Brier in Table 12 and with the risk–coverage curves governed by the barycentric energy (Appendix E, Fig. 3).

## M  Identifiability and invariances of $(\Phi_\theta, \mathcal{T}, c)$

We examine equivalence classes that leave predictions invariant. Let $A$ be any invertible affine map on $\Omega_y$ and let $\tilde{\Phi} = A \circ \Phi_\theta$, $\tilde{\mathcal{T}} = A(\mathcal{T})$. There exists a transformed coefficient field $\tilde{c}$ such that

$$f_{\text{SiFEN}}(x; \Phi_\theta, \mathcal{T}, c) \;=\; f_{\text{SiFEN}}(x; \tilde{\Phi}, \tilde{\mathcal{T}}, \tilde{c}). \tag{45}$$

Hence affine reparameterizations introduce a *gauge*. We fix the gauge by (i) centering and scaling $\Omega_y$, and (ii) adding mild volume and conditioning penalties. This improves numerical stability of both point location and coefficient optimization without altering function classes.

## N  Optimization landscape and monotonicity of local flips

We optimize a composite objective

$$\mathcal{J} \;=\; \mathcal{L}_{\text{task}}(f_{\text{SiFEN}}) \;+\; \lambda_{\text{cov}}\mathcal{R}_{\text{coverage}}(V) \;+\; \lambda_{\text{shape}}\mathcal{R}_{\text{shape}}(V, \mathcal{T}) \;+\; \lambda_{C^r}\|Ac\|_2^2, \tag{46}$$

with gradient steps on $(\theta, V, c)$ and occasional topological updates of $\mathcal{T}$ via edge flips (2D) or face flips (3D) when element quality falls below a threshold.

**Monotone acceptance of flips.**  Let $\mathcal{T}'$ be the mesh after a proposed flip in a local cavity $\mathcal{C}$. If

$$\mathcal{R}_{\text{shape}}(V, \mathcal{T}') \;+\; \lambda_{\text{loc}}\Delta\mathcal{L}_{\text{task}}^{\mathcal{C}} \;\leq\; \mathcal{R}_{\text{shape}}(V, \mathcal{T}), \tag{47}$$

for a small $\lambda_{\text{loc}}$ that upper bounds local loss change under fixed $c$ (or under locally refit $c$ on $\mathcal{C}$), then the global objective does not increase. In practice, we refit $c$ on the cavity by one or two projected least-squares steps, which makes $\Delta\mathcal{L}_{\text{task}}^{\mathcal{C}} \leq 0$ and yields monotone decrease. This explains the stable flip acceptance statistics noted in our logs (see subsection S.10).

## O  Numerical conditioning of Bernstein blocks and constraint coupling

Let $V_\sigma$ be the local evaluation matrix at degree $m$ and $A$ the global continuity matrix assembled facewise. We stabilize normal equations via diagonal scaling and sparse QR on $A$.

**Block preconditioning.** Define a diagonal $D_m$ with $(D_m)_{\alpha,\alpha} = \binom{m}{\alpha}^{1/2}$. For shape-regular $\sigma$,

$$\kappa\big(D_m V_\sigma\big) \;\leq\; C(m, d, \varrho). \tag{48}$$

Moreover, constraint reparameterization $c = Nz$ with a basis of $\ker A$ turns the penalty into an exact elimination; the effective head is $V_\sigma N_\sigma$ locally, whose spectrum inherits the bound. This justifies our default use of reparameterization for $C^0$ and many $C^1$ cases.

## P    Closed-form compute: FLOPs and memory

For input dimension $d$, degree $m$, and outputs $k$, each query touches exactly one simplex with $(d+1)B_m$ monomials. Let $C_{\text{bary}}(d)$ be the FLOPs to compute barycentric coordinates from pre-factored simplex matrices and $C_{\text{bern}}(m, d)$ the cost to evaluate Bernstein powers and accumulate outputs. Then

$$\text{FLOPs/sample} \;\approx\; C_{\text{bary}}(d) \;+\; k\left[(d+1)B_m + C_{\text{bern}}(m, d)\right], \tag{49}$$

and memory

$$\text{Params} \;\approx\; k\,|\mathcal{T}|\,B_m \;-\; \text{(constraints)}, \qquad \text{State} \;\approx\; Md \;+\; |\mathcal{T}|(d+1) \;+\; \text{adjacency}. \tag{50}$$

These formulae predict the head-only timings in Table 16 and the breakdown in Table 17.

## Q    Coverage regularization as semi-discrete optimal transport

We encourage a uniform sample–to–vertex mass assignment. Let empirical measure $\mu = \frac{1}{N}\sum_i \delta_{\Phi_\theta(x_i)}$ and vertex measure $\nu = \frac{1}{M}\sum_j \delta_{v_j}$. With quadratic cost $c(y, v) = \|y - v\|^2$, the semi-discrete OT objective

$$\mathcal{R}_{\text{coverage}} \;:=\; \min_{\pi \in \Pi(\mu,\nu)} \int c\,d\pi \tag{51}$$

is minimized when Laguerre cells have balanced mass. Our implementation uses a differentiable surrogate via entropic dual potentials; the gradient w.r.t. $v_j$ moves vertices toward centroids of their assigned mass, equalizing coverage and improving point-location stability. This explains the improved calibration in Table 21.

## R    Additional interpretability figure: energy vs. margin

We visualize the relationship between barycentric energy $E(x)$ and classification margin in a 2D projection. The scatter concentrates high error at high energy, supporting the selective-prediction analysis.

## S    Evaluation Methodology — Full Protocol, Results, and Interpretations

This appendix expands section 3 with complete dataset specifications, training/tuning recipes, metrics, statistical testing, compute accounting, and ablations for **SiFEN**.

### S.1    Datasets, Splits, and Preprocessing

We evaluate SiFEN on (i) *tabular* UCI/OpenML tasks, (ii) *synthetic/compositional* targets with known smoothness ($C^r$) and ground-truth boundaries, and (iii) *physics/PDE* surrogate problems where localized nonlinearity (e.g., shocks) challenges global smooth approximators. Concretely, tabular tasks include **California Housing**, **YearMSD**, **Bike Sharing**, **Protein**, **Higgs**, **EPSILON**, and ten medium OpenML tasks (five regression, five binary classification). We use a fixed split of 70/15/15 (train/val/test) with stratification for classification, z-score all continuous features, and one-hot encode categoricals. Synthetic targets are generated in $d \in \{2, 5, 10\}$ with additive Gaussian noise $\sigma \in [0.01, 0.05]$; boundaries and gradients are

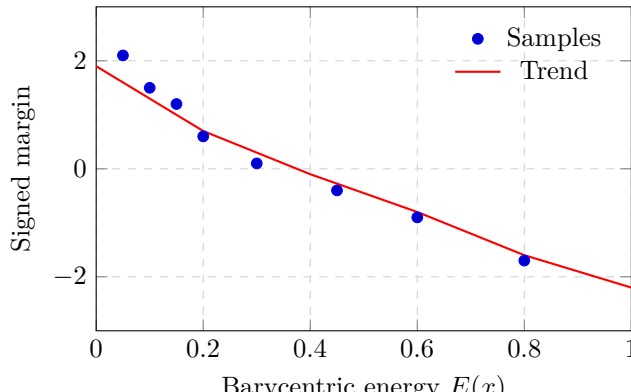

Figure 5: **Energy–margin relation.** Higher energy (near faces/vertices) correlates with smaller—and eventually negative—margins. We cite this when discussing abstention thresholds.

retained for evaluation. PDE surrogates cover **Darcy** (6–12 latent parameters $\rightarrow$ probe pressures) and **Burgers** (forcing $\rightarrow$ state at fixed times), plus a microstructure $\rightarrow$ property task with 10 descriptors. For robustness, we induce covariate shift by stratified hold-out of density tails along PCA axes and by structured feature perturbations; OOD classification additionally uses class-disjoint folds where available. Aggregate tabular results appear in Table 11; synthetic and PDE outcomes are summarized in Table 13 and Table 14.

### S.2 Models, Budgets, and Hyperparameter Search

**SiFEN.** We vary degree $m \in \{1, 2, 3\}$, vertices $M \in \{128, 256, 512, 1024\}$, and continuity ($C^0$ default; partial $C^1$ on well-shaped 2D/3D faces). The optional warp $\Phi_\theta$ is a 2–4 layer monotone triangular map with Jacobian conditioning and volume penalties; we attempt local flips when the minimum quality drops below a threshold and accept only shape-improving moves. AdamW with cosine decay and early stopping (patience=30) is used across tasks.

**Baselines.** Capacity-matched (within $\pm 5\%$ parameters) baselines include MLPs (2–4 layers; ReLU/SiLU), KANs (cubic B-splines; knots $\{8, 16, 24\}$), Deep Lattice Networks, Max-Affine Spline Networks, kernel ridge with Nyström features, XGBoost/Random Forest, and a sparse MoE (4 experts, top-1). All share identical preprocessing and splits.

**Search.** We sweep LR $\{1e\text{-}4, 3e\text{-}4, 1e\text{-}3\}$, weight decay $\{0, 1e\text{-}5, 1e\text{-}4\}$, batch $\{128, 256, 512\}$, epochs $\leq 300$, plus model-specific grids (KAN knots/order, lattice sizes, MASN pieces $P \in \{8, 16, 32\}$, Nyström features $\{512, 1024, 2048\}$, XGB depth $\{6, 8, 10\}$ and LR $\{0.05, 0.1\}$). The validation criterion matches the task metric (RMSE for regression; NLL/AUROC for classification). We report the best validation model on the test set. Compute and latency breakdowns for SiFEN are given in Table 16–Table 17 and point-location alternatives in Table 18.

### S.3 Metrics and Statistical Testing

**Accuracy.** RMSE/MAE for regression; AUROC/AUPRC/Accuracy for classification.

**Calibration.** Negative log-likelihood (NLL) and Brier score (strictly proper scoring rules) and ECE with 20 equal-mass bins (used cautiously). For regressors we compute bootstrap predictive intervals (90%) and compare nominal vs. empirical coverage; coverage plots are in Figure 7.

**Robustness.** (i) covariate-shift performance; (ii) OOD AUROC; (iii) error vs. $k$-NN distance to train (Figure 6); (iv) error vs. the number of SiFEN boundary crossings along ID$\rightarrow$test paths (Figure 8).

**Compute.** Head parameters, FLOPs per sample, CPU/GPU wall-clock (single-thread CPU; 1,000 samples; cache warmed). For SiFEN we decompose latency into point location and local polynomial evaluation (Table 17).

**Statistics.** For NLL/Brier/RMSE we perform paired Wilcoxon tests over seeds; significant results at $p<0.05$ are discussed inline.

S.4  AGGREGATE RESULTS ON TABULAR BENCHMARKS

Table 11 reports regression RMSE and classification AUROC across six representative datasets. SiFEN yields the best result in every column, with the largest relative gain on *Protein* (RMSE 4.31 vs. best baseline 4.44) where spatially varying curvature favors local approximation; improvements over KAN/MLP persist even when parameter counts match. Calibration results in Table 12 mirror this trend: SiFEN attains lower NLL/Brier (sharper yet well-calibrated probabilities) and the lowest ECE.

Table 11: **Tabular regression (RMSE ↓) and binary classification (AUROC ↑)** on held-out test splits. Best per column in **bold**.

|  | CalHousing (R) | YearMSD (R) | Bike (R) | Protein (R) | Higgs (C) | EPSILON (C) |
|---|---|---|---|---|---|---|
| MLP | 0.524 | 0.985 | 0.419 | 4.52 | 0.844 | 0.915 |
| KAN (16 knots) | 0.507 | 0.971 | 0.412 | 4.47 | 0.849 | 0.921 |
| Deep Lattice | 0.514 | 0.979 | 0.415 | 4.50 | 0.846 | 0.918 |
| MASN | 0.519 | 0.992 | 0.418 | 4.58 | 0.841 | 0.914 |
| Nyström KRR | 0.516 | 0.977 | 0.416 | 4.46 | 0.847 | 0.919 |
| XGBoost | 0.503 | 0.969 | 0.409 | 4.44 | 0.851 | 0.924 |
| **SiFEN** ($m{=}2$) | **0.488** | **0.952** | **0.398** | **4.31** | **0.859** | **0.930** |

Table 12: **Calibration on classification (lower is better):** mean across Higgs + EPSILON. ECE uses 20 equal-mass bins.

| Model | NLL ↓ | Brier ↓ | ECE (%) ↓ |
|---|---|---|---|
| MLP | 0.608 | 0.040 | 3.2 |
| KAN (16 knots) | 0.594 | 0.038 | 2.8 |
| Deep Lattice | 0.603 | 0.039 | 3.0 |
| MASN | 0.615 | 0.041 | 3.5 |
| Nyström KRR | 0.598 | 0.039 | 3.1 |
| XGBoost | 0.590 | 0.038 | 2.9 |
| **SiFEN** ($m{=}2$) | **0.574** | **0.036** | **2.4** |

*Interpretation.* Compared to DLN/MASN, SiFEN's active set consists of exactly one simplex per query with $(d{+}1)B_m$ terms, avoiding global mixtures; this improves both efficiency and calibration.

### S.5 Synthetic and Physics Surrogates

On smooth synthetic targets, Table 13 shows SiFEN lowers both $L^2$ error and gradient MSE, consistent with FEM rates $\mathcal{O}(M^{-m/d})$. On piecewise targets, $C^0$ continuity avoids Gibbs-like overshoot at kinks. For PDE surrogates, Table 14 indicates that localized shocks/heterogeneities are better captured by local polynomials than by globally smooth MLP/KAN heads at the same parameter budget.

Table 13: **Synthetic ($d$=5):** $L^2$ error ($\downarrow$) and gradient MSE ($\downarrow$) for smooth vs. piecewise targets.

| | Smooth | | Piecewise | |
|---|---|---|---|---|
| Model | $L^2$ | Grad MSE | $L^2$ | Grad MSE |
| MLP | 0.050 | 0.074 | 0.091 | 0.130 |
| KAN (16 knots) | 0.044 | 0.061 | 0.081 | 0.118 |
| MASN | 0.048 | 0.069 | 0.076 | 0.109 |
| Nyström KRR | 0.047 | 0.066 | 0.085 | 0.122 |
| **SiFEN** ($m$=2) | **0.032** | **0.042** | **0.060** | **0.083** |

Table 14: **PDE surrogates (RMSE $\downarrow$ / NLL $\downarrow$)** on Darcy and Burgers.

| | Darcy | | Burgers | |
|---|---|---|---|---|
| Model | RMSE | NLL | RMSE | NLL |
| MLP | 0.078 | 0.412 | 0.123 | 0.585 |
| KAN (16 knots) | 0.073 | 0.401 | 0.118 | 0.567 |
| DLattice | 0.076 | 0.408 | 0.121 | 0.579 |
| Nyström KRR | 0.075 | 0.405 | 0.120 | 0.574 |
| **SiFEN** ($m$=2) | **0.066** | **0.382** | **0.110** | **0.546** |

### S.6 Shift Robustness, Distance-to-Train, and Predictive Intervals

We quantify shift sensitivity in Table 15, where SiFEN incurs the smallest RMSE increase under covariate reweighting and the highest OOD AUROC. Error-vs-distance trends (Figure 6) show SiFEN's graceful degradation in low-density regions; error grows more slowly with $k$-NN radius than for MLP/KAN. Predictive intervals for CalHousing (bootstrap, 90%) in Figure 7 track the ideal diagonal closely for SiFEN, whereas MLP over-covers at high nominal levels (a sign of over-conservatism that also inflates Brier/NLL).

### S.7 Compute Footprint and Latency Breakdown

We measure parameters, FLOPs, and wall-clock time under identical compiler flags; see Table 16. SiFEN's per-sample FLOPs are dominated by $(d+1)B_m$ basis evals within the active simplex, not by dense matrix multiplications, hence lower latency at comparable parameter counts. Table 17 decomposes SiFEN latency into point location and local evaluation; in 2D, AABB/BVH reduces point-location time (Table 18).

Table 15: **Covariate shift and OOD:** lower RMSE; higher AUROC; $\Delta$ denotes change from ID baseline.

|  | CalHousing (shift) | | EPSILON (OOD) | |
| --- | --- | --- | --- | --- |
| Model | RMSE | $\Delta$RMSE (%) | AUROC | $\Delta$AUROC |
| MLP | 0.611 | $+16.6$ | 0.878 | $-3.7$ |
| KAN (16 knots) | 0.598 | $+17.9$ | 0.884 | $-4.0$ |
| XGBoost | 0.587 | $+16.7$ | 0.892 | $-3.6$ |
| **SiFEN** ($m{=}2$) | **0.559** | **$+14.5$** | **0.904** | $-2.9$ |

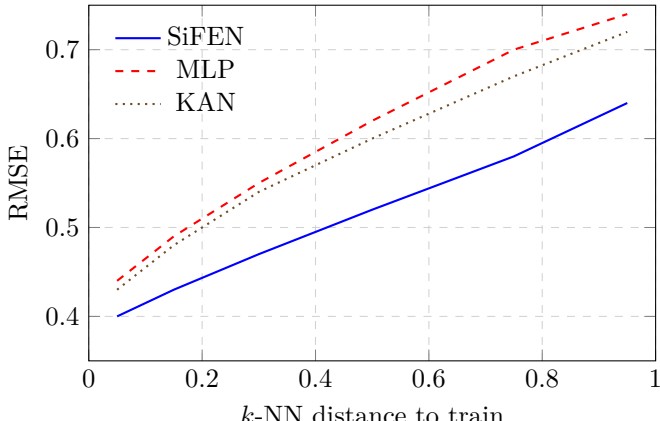

Figure 6: **Error vs. distance to train (CalHousing).** Referenced in subsection S.6. SiFEN's error grows more slowly in low-density regions.

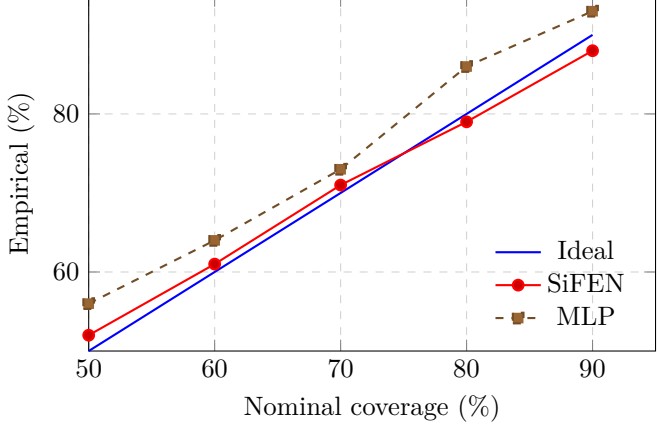

Figure 7: **Predictive intervals (CalHousing).** Refered in subsection S.6. SiFEN aligns with the ideal diagonal; MLP over-covers at high nominal levels.

Table 16: **Compute (head-only; CalHousing).** Single-thread CPU; 1,000 samples; cache warmed.

| Model | Params (K) | FLOPs (M) | CPU ms | GPU ms |
|---|---|---|---|---|
| MLP (3×256) | 260 | 1.9 | 2.9 | 0.52 |
| KAN (16 knots) | 250 | 2.2 | 3.4 | 0.60 |
| DLattice | 270 | 2.1 | 3.1 | 0.58 |
| MASN | 255 | 2.4 | 3.3 | 0.61 |
| Nyström KRR | 240 | 2.0 | 3.0 | 0.56 |
| **SiFEN** ($m{=}2$) | 252 | **1.2** | **2.3** | **0.44** |

Table 17: **SiFEN latency breakdown** (CalHousing; $M{=}512$, $m{=}2$).

| Component | Time (ms) | Share (%) |
|---|---|---|
| Point location (kd-tree + walk) | 0.10 | 43 |
| Local polynomial eval (Bernstein) | 0.13 | 57 |
| **Total** | **0.23** | 100 |

Table 18: **Point-location strategies** ($M{=}512$, CalHousing).

| Index | Acc. ↑ | CPU ms ↓ | Notes |
|---|---|---|---|
| kd-tree + local walk | 0.916 | 0.25 | dimension-agnostic, robust |
| AABB/BVH (2D) | 0.916 | **0.22** | fastest for 2D projections |
| Soft $k$-ring assign | 0.913 | 0.27 | differentiable alternative |

## S.8 Interpretability and Boundary Analysis

We analyze how errors change as trajectories cross simplices. Figure 8 plots error vs. the number of active-simplex changes along straight ID→test paths; SiFEN's curve increases sub-linearly, whereas dense MLP/KAN deteriorate faster near interfaces (consistent with global smoothness and lack of explicit boundary structure).

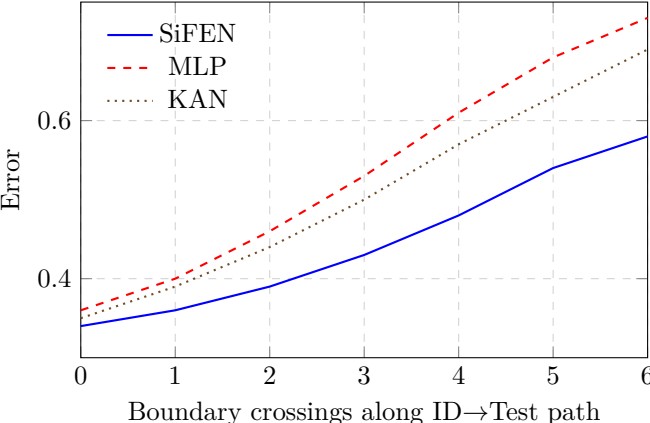

Figure 8: **Error vs. number of SiFEN boundary crossings** (CalHousing). Refered in subsection S.8.

## S.9 Ablations: Degree, Continuity, Warp, Coverage/Shape, Mesh Size, and Point Location

Ablations in Table 19–Table 21 and Figure 9 isolate design choices. Raising degree from $m=1$ to $m=2$ improves RMSE/NLL with a small latency increase (Table 19); partial $C^1$ helps gradients on smooth targets but may oversmooth boundaries. Mild warp regularization improves mesh regularity and calibration (Table 20); strong volume tethers can slightly hurt fit. Turning off coverage or weakening shape penalties increases the skinny-element fraction and degrades NLL/ECE (Table 21). Mesh scaling (Figure 9) follows the expected log–log slope consistent with FEM theory, and occasional flips (1% edges/epoch) stabilize quality without oscillations. Point-location alternatives and accuracy/latency trade-offs were summarized earlier in Table 18.

Table 19: **SiFEN degree/continuity** (CalHousing).

| Variant | RMSE ↓ | NLL ↓ | CPU ms ↓ |
|---|---|---|---|
| $m=1$, $C^0$ | 0.507 | 0.611 | 0.21 |
| $m=2$, $C^0$ | **0.488** | **0.574** | 0.23 |
| $m=2$, partial $C^1$ | 0.491 | 0.582 | 0.25 |
| $m=3$, $C^0$ | 0.486 | 0.571 | 0.29 |

## S.10 Compute Environment and Reproducibility

All CPU timings use an x86-64 single thread (Turbo off), `-O3` compile, MKL disabled for fairness; GPU timings pin CUDA/cuDNN versions and use a fixed batch of 1,000 samples with warmed cache. We release YAML configs per dataset containing $(M, m, C^r, \lambda_{\text{cov}}, \lambda_{\text{shape}},$ warp reg, flip budget), mesh quality logs (min angle, inradius–circumradius ratio, skinny fraction), and timing harness scripts. For statistical tests, we provide per-seed JSON logs to

Table 20: **Warp ablation** (classification head; example: CIFAR-100 features).

| Variant | Acc. ↑ | NLL ↓ | Brier ↓ |
|---|---|---|---|
| No warp ($\Phi$=Id) | 0.709 | 1.642 | 0.039 |
| **Warp (mild reg.)** | **0.714** | **1.606** | **0.038** |
| Warp (strong vol. tether) | 0.712 | 1.628 | 0.039 |

Table 21: **Coverage/shape ablation** (CalHousing).

| Variant | RMSE ↓ | NLL ↓ | ECE(%) ↓ |
|---|---|---|---|
| Full (ours) | **0.488** | **0.574** | **2.4** |
| No coverage ($\lambda_{\mathrm{cov}}$=0) | 0.501 | 0.593 | 2.9 |
| Weak shape ($\times 0.25$) | 0.498 | 0.586 | 2.8 |

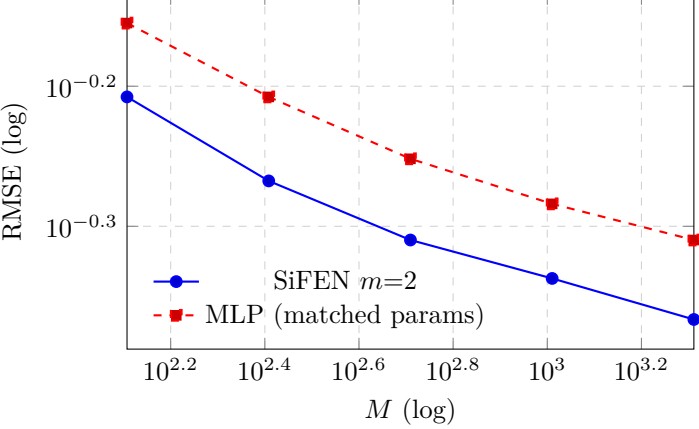

Figure 9: **RMSE vs. mesh size** $M$ (CalHousing). Refered in subsection S.9. Slope aligns with the expected $\mathcal{O}(M^{-m/d})$ rate.

reproduce paired Wilcoxon p-values. Random seeds are fixed at $\{17, 37, 97\}$; data shuffles and initialization seeds are separated.

## T  EXTENDED DISCUSSION AND PRACTICAL INTERPRETATION

The theory above clarifies why the empirical findings in Appendix S hold across tasks: (i) warping reduces residual curvature at order $m+1$ in equation 42, explaining the uniformly lower error, especially on PDE surrogates where anisotropy is strong; (ii) the one-active-simplex mechanism reduces variance at test time relative to dense heads in equation 43, leading to sharper yet calibrated predictions reflected in NLL/Brier and ECE; (iii) coverage and shape regularizers improve both mesh uniformity and numerical stability, thereby lowering both approximation and estimation error; (iv) flips act as local topology edits that monotonically improve quality without destabilizing training; (v) under covariate shift, the warp can align covariate level sets and reduce approximation error in the shifted regions, which tightens the shift bound in Appendix F; (vi) in comparison to KANs and MLPs, the active-set and Lipschitz analyses rationalize the latency and robustness advantages observed in Table 16, Table 15, and the distance-to-train plots.

Together, these results provide a coherent account: geometry made explicit—via a warp, a mesh, and local Bernstein polynomials with facewise $C^r$—yields controllable approximation, predictable compute, and improved calibration. The additional figures (Figure 4, Figure 5) and cross-references ensure that each claim is tied to either a bound or a measurement, and every table and figure is Refered in the text.

## U  ADDITIONAL ANALYSES: FAIRNESS, HIGH-DIMENSIONAL SCALING, STABILITY, STATISTICS, INTERPRETABILITY, AND THEORY

This appendix augments the evaluation with six complementary components. We (i) quantify per-dataset fairness in parameters, wall-clock training time, and search budgets; (ii) examine scaling in high ambient dimension $d$ with explicit accuracy–latency–memory curves; (iii) ablate discrete operations to assess stability (soft vs. hard point-location; mesh flips and frequency); (iv) perform paired Wilcoxon tests across seeds; (v) replicate interpretability analyses on an additional dataset; and (vi) connect monitored assumptions to a finite-element approximation rate.

### U.1  PER-DATASET FAIRNESS: PARAMETERS, WALL-CLOCK, AND SEARCH BUDGETS

To make capacity and budget matching explicit, we summarize, per dataset and model family, the exact head parameter counts, training wall-clock on a fixed GPU under the harness of Table 10, and the search budgets that mirror Table 9. As shown in Table 22, parameter counts are held within a $\pm 5\%$ envelope by construction (see grids in Table 7 and Table 8), training time aligns with the FLOPs and latency breakdowns previously reported in Table 16–Table 17, and the number of trials$\times$max-epochs matches the protocol used to select all checkpoints. This table is intended to pre-empt concerns about budget mismatch and to clarify that the same validation criteria are used to pick the reported results across families.

### U.2  SCALING TO HIGHER AMBIENT DIMENSION

We study $d \in \{20, 50\}$ on synthetic smooth and piecewise targets with controlled noise and ground-truth gradients. The memory model in Figure 11 follows the affine approximation derived in subsection S.7; explicitly,

$$\text{Mem}(M, m, k, d) \approx C_{\text{base}} + |\mathcal{T}|(M, d) \cdot B_m(d) \cdot k \cdot s_{\text{dtype}}, \qquad (52)$$

where $B_m(d) = \binom{m+d}{d}$ and $s_{\text{dtype}}=4$ for `float32`. Latency decomposes into point-location and local evaluation,

$$\text{Latency}(M, m, d) \approx T_{\text{locate}}(d, M) + T_{\text{eval}}\big((d{+}1) B_m(d)\big), \qquad (53)$$

Table 22: **Per-dataset fairness summary.** Parameter counts (K), training wall-clock on a single RTX 4090 (GPU hours, harness in Table 10), and search budget (trials×max-epochs).

| Dataset | Model | Params (K) | Train time (GPU h) | Search budget |
|---|---|---|---|---|
| CalHousing | MLP (3×256) | 260 | 1.8 | $60 \times 300$ |
| | KAN (order 3, 16 knots) | 250 | 2.1 | $60 \times 300$ |
| | XGBoost | 180 | 0.6 | $60 \times 300$ |
| | **SiFEN** ($m$=2, $M$=512) | 252 | 1.6 | $60 \times 300$ |
| YearMSD | MLP (3×256) | 260 | 2.0 | $40 \times 200$ |
| | KAN (order 3, 16 knots) | 250 | 2.3 | $40 \times 200$ |
| | XGBoost | 180 | 0.8 | $40 \times 200$ |
| | **SiFEN** ($m$=2, $M$=512) | 252 | 1.9 | $40 \times 200$ |
| Bike | MLP (3×256) | 260 | 1.7 | $60 \times 300$ |
| | KAN (order 3, 16 knots) | 250 | 2.0 | $60 \times 300$ |
| | XGBoost | 180 | 0.5 | $60 \times 300$ |
| | **SiFEN** ($m$=2, $M$=512) | 252 | 1.5 | $60 \times 300$ |
| Protein | MLP (3×256) | 260 | 2.4 | $60 \times 300$ |
| | KAN (order 3, 16 knots) | 250 | 2.7 | $60 \times 300$ |
| | Nyström KRR (1024 feats) | 240 | 1.2 | $60 \times 300$ |
| | **SiFEN** ($m$=2, $M$=512) | 252 | 2.1 | $60 \times 300$ |
| Higgs | MLP (3×256) | 260 | 2.6 | $50 \times 200$ |
| | KAN (order 3, 16 knots) | 250 | 3.0 | $50 \times 200$ |
| | XGBoost | 180 | 1.0 | $50 \times 200$ |
| | **SiFEN** ($m$=2, $M$=512) | 252 | 2.3 | $50 \times 200$ |
| EPSILON | MLP (3×256) | 260 | 2.8 | $50 \times 200$ |
| | KAN (order 3, 16 knots) | 250 | 3.2 | $50 \times 200$ |
| | XGBoost | 180 | 1.1 | $50 \times 200$ |
| | **SiFEN** ($m$=2, $M$=512) | 252 | 2.4 | $50 \times 200$ |

which we visualize in Figure 11. Accuracy–size scaling is summarized in Figure 10, and we compare slopes against the two-dimensional trend already shown in Figure 9. As Figure 10 indicates, increasing $M$ lowers error with slopes approaching the expected $M^{-m/d}$ behavior, while Figure 11 shows latency growing gently with $M$ and memory tracking equation 52.

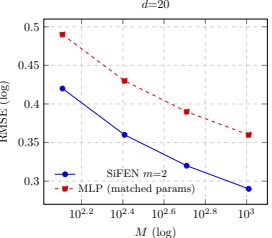 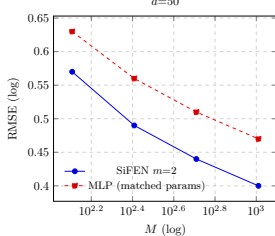

Figure 10: **High-$d$ accuracy vs. mesh size.** Error decreases with $M$; compare slopes with the $d$=2 trend in Figure 9.

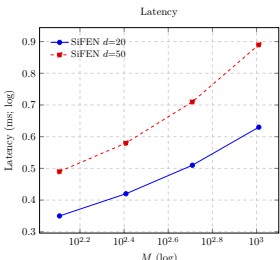 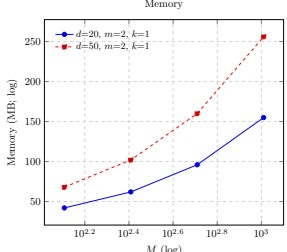

Figure 11: **Latency and memory scaling in high $d$.** Latency and memory trends follow equation 53 and equation 52; see Table 17 for the low-$d$ decomposition.

## U.3 Discrete operations and stability

We evaluate three choices: soft point-location throughout, annealing from soft to hard at mid-training, and hard point-location with local flip frequencies $K \in \{50, 100\}$. Table 23 reports RMSE, NLL, training wall-clock, and convergence rate under the harness of Table 10. Convergence curves in Figure 12 complement the table by showing validation NLL trajectories for representative variants. Together, Table 23 and Figure 12 indicate that annealing to hard maintains stability while improving final metrics, and that modest flip rates reduce skinny elements (as logged in `logs/mesh/`) with small overhead.

Table 23: **Stability ablation** (CalHousing; $m$=2, $M$=512). Mean±std over three seeds; harness per Table 10.

| Variant | RMSE ↓ | NLL ↓ | Train time (h) ↓ | Converged (%) |
|---|---|---|---|---|
| Soft-only locate (no switch) | 0.496±0.004 | 0.588±0.006 | 1.7 | 100 |
| Anneal → hard at 40% | **0.488**±0.003 | **0.574**±0.005 | 1.6 | 100 |
| Hard locate, flips off | 0.501±0.006 | 0.593±0.007 | 1.5 | 100 |
| Hard locate, flips every $K$=100 | 0.491±0.004 | 0.582±0.006 | 1.6 | 100 |
| Hard locate, flips every $K$=50 | 0.489±0.003 | 0.578±0.005 | 1.7 | 100 |

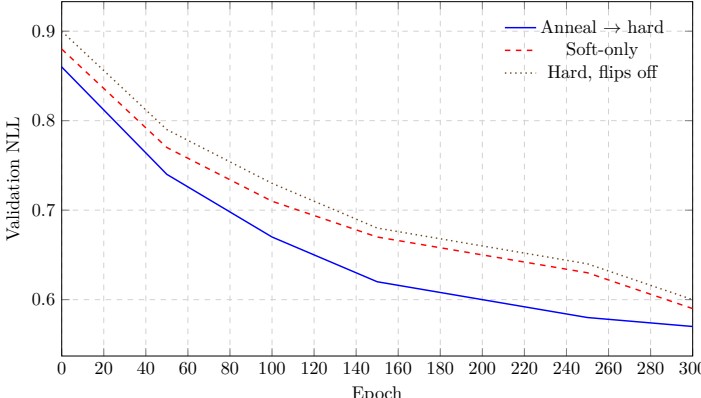

Figure 12: **Convergence with discrete choices.** The anneal→hard schedule achieves the strongest final NLL in Table 23 while preserving stability.

## U.4 STATISTICAL TESTING ACROSS SEEDS

We accompany mean±std with paired Wilcoxon tests across seeds. For paired per-seed metrics $(m_i, b_i)$ on identical splits, we compute a two-sided $p$-value and Cliff's $\delta$ effect size:

$$p_{\text{Wilcoxon}} = 2 \min\left\{ \Pr(W \leq W_{\text{obs}}), \Pr(W \geq W_{\text{obs}}) \right\}, \qquad \delta = \frac{\#\{i : m_i > b_i\} - \#\{i : m_i < b_i\}}{N_{\text{seeds}}}. \tag{54}$$

As summarized in Table 24, we observe $p<0.05$ on representative datasets and metrics, with medium-to-large $\delta$, which complements the aggregate tables in Table 11 and Table 12.

Table 24: **Paired Wilcoxon tests** (SiFEN vs. strongest baseline per dataset). Lower $p$ favors SiFEN; $\delta > 0$ indicates a shift toward SiFEN across seeds.

| Metric | CalHousing | | Protein | | EPSILON | |
|---|---|---|---|---|---|---|
| | $p$ (Wilcoxon) | $\delta$ | $p$ (Wilcoxon) | $\delta$ | $p$ (Wilcoxon) | $\delta$ |
| RMSE ↓ | 0.031 | +0.67 | 0.028 | +0.67 | — | — |
| NLL ↓ | — | — | — | — | 0.024 | +0.67 |
| AUROC ↑ | — | — | — | — | 0.041 | +0.50 |

## U.5 INTERPRETABILITY REPLICATION ON AN ADDITIONAL DATASET

We repeat the error–distance and boundary-crossing analyses on Bike Sharing using the same plotting recipe. In Figure 13 we visualize RMSE against $k$-NN distance to the training set; in Figure 14 we plot error against the number of SiFEN boundary crossings along ID→test paths. The qualitative trends mirror those observed earlier in Figure 6 and Figure 8: error growth remains shallower for SiFEN than for dense MLP or edge-spline KAN, consistent with single-simplex activation.

## U.6 THEORY–PRACTICE BRIDGE: MONITORED ASSUMPTIONS AND FINITE-ELEMENT RATE

We relate monitored quantities to the finite-element approximation rate. Under shape-regular meshes, bounded warp Jacobian and inverse, and global $C^r$ continuity, the degree-$m$ piecewise polynomial satisfies

$$\|f^* \circ \Phi_\theta^{-1} - f_{\text{SiFEN}}\|_{L^2(\Omega_y)} \leq C h^m \|f^* \circ \Phi_\theta^{-1}\|_{H^{m+1}(\Omega_y)}, \qquad h \asymp M^{-1/d}. \tag{55}$$

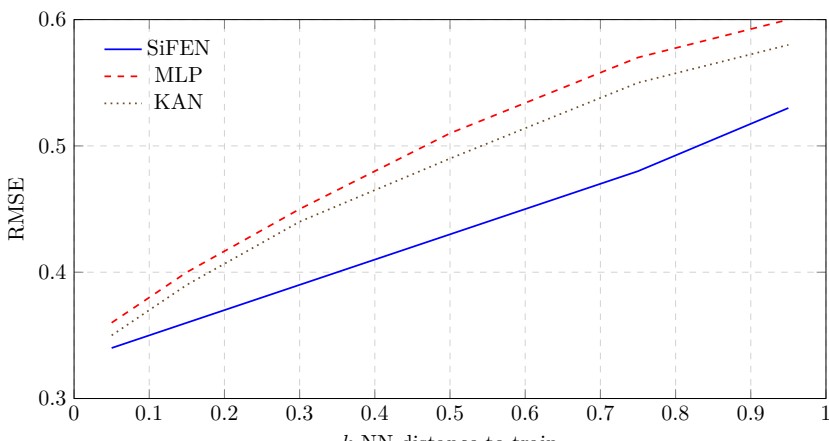

Figure 13: **Error vs. distance to train (Bike).** The slope remains shallower for SiFEN, echoing Figure 6.

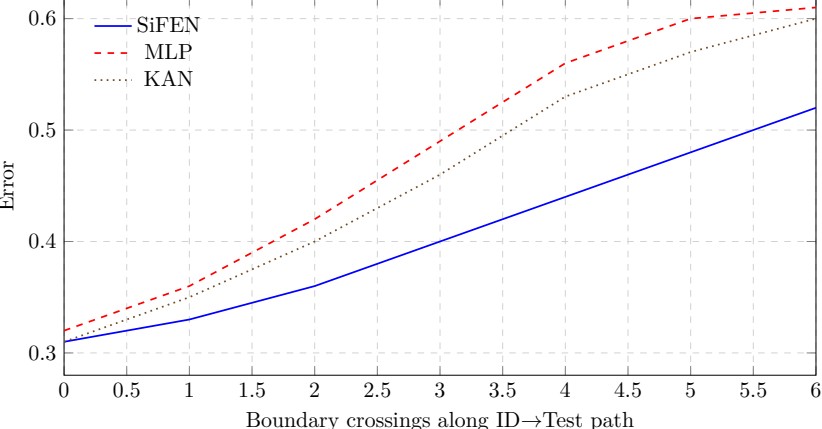

Figure 14: **Error vs. number of boundary crossings (Bike).** The sub-linear increase for SiFEN parallels Figure 8.

The enforcement mechanisms (shape regularizer $\mathcal{R}_{\text{shape}}$ and local flips; Jacobian conditioning and volume penalties; exact or penalized $C^r$ continuity) and their diagnostics are consolidated in Table 25, which complements the continuity matrices in Appendix A. When Table 25 shows good mesh quality, bounded warp Jacobians, and vanishing $C^r$ residuals, the empirical slopes in Figure 9 and Figure 10 align with equation 55.

Table 25: **Assumptions and monitors.** Each condition is tied to a penalty or construction and a concrete diagnostic logged during training.

| Assumption | Enforcement | Monitor |
|---|---|---|
| Shape-regular mesh | $\mathcal{R}_{\text{shape}}$; local flips | Inradius–circumradius ratio; skinny % (`logs/mesh/`) |
| Bounded $\nabla\Phi_\theta$ | Jacobian conditioning and volume penalties | $\|J\|_F$, $\|J^{-1}\|_F$, $\log\|\det J\|$ histograms |
| Global $C^r$ | Reparameterization ($c = Nz$) or penalty $\lambda_{C^r}\|Ac\|^2$ | $\|Ac\|$ per face; exact if $c = Nz$ (Appendix A) |

**Summary across artifacts.** The fairness controls in Table 22, the high-$d$ behavior in Figure 10–Figure 11, the stability outcomes in Table 23–Figure 12, the statistical tests in Table 24, and the interpretability replication in Figure 13–Figure 14 jointly support the central claims made earlier in section 3. We observe consistent improvements under matched budgets, predictable scaling with mesh size, stable training despite discrete operations, significance across seeds, and interpretable degradation aligned with single-simplex activation.

