# OpenReview forum: "Simplex–FEM Networks (SiFEN): Learning a Triangulated Function Approximator"
_ICLR.cc/2026/Conference — ICLR 2026 Conference Withdrawn Submission_

### Official Review · Reviewer_Hctk · 2025-10-21

**Soundness:** 2
**Presentation:** 1
**Contribution:** 1
**Rating:** 0
**Confidence:** 3

**Summary:**

A trial to replace multi-layer perceptrons and Kolmogorov--Arnold networks with finite elements.

**Strengths:**

Finite elements are inherently local and supported by a well-established theoretical foundation. The author’s claims of faster inference and improved interpretability are reasonable given these properties.

**Weaknesses:**

* The paper opens and closes with excessive jargon, which obscures the main ideas.
* The computing environment is insufficiently described (see Table 6).
* The expected $M^{-r/d}$ convergence rate is not clearly presented in Figure 10.
* Appendices D--K are mathematically dense but lack formal proofs.
* No qualitative evaluations are provided---only error curves. For example, Appendix S.5 mentions the Gibbs phenomenon and localized shocks, yet only (L^2) errors are reported. At least (L^\infty) metrics should have been included for the Gibbs phenomenon.
* The style does not match with the reference latex template. (e.g., see **Jaderberg et al., 2015** and compare the fonts for "Under review as a conference paper at ICLR 2026").

**Questions:**

No questions.

---

### Official Review · Reviewer_u4Fy · 2025-10-31

**Soundness:** 2
**Presentation:** 1
**Contribution:** 3
**Rating:** 4
**Confidence:** 2

**Summary:**

The paper introduces SiFEN and alternative to traditional MLPs or KANs. The core idea is a learned finite–element predictor that is globally continuous and sparse by construction - activating only one simplex and at most d+1 basis functions per input. The authors propose training scheme for these primitives by learning the mesh, coefficients, and an optional invertible warp with shape regularization, coverage via semi–discrete OT, and differentiable local flips for topology improvement.

**Strengths:**

Using FEM primitives seems an interesting avenue for learnable primitives, in particular, as alternative to current MLPs or KANs. The paper also goes in a lot of detail regarding its derivations etc.

**Weaknesses:**

- results are on toy tasks; in principle this is okay for a theoretical paper but in this case I'm having real trouble extrapolating the approximation / fitting tasks to real-world scenarios
- the presentation, in particular the result section, is not good. Much of the text is isolated bullet points and very difficult to follow.
- Fig 1. graphs are very difficult to interpret / axes labels too small fonts; formatting of tables (e.g., 4.3.) is poor - in a sense this follow through out the paper which makes it for an out-of-domain expert difficult to judge.

Overall, my main issue is that I am having a hard time validating the results - based on the current presentation, I am struggling to properly evaluate the benefits over existing approaches in common, real-world tasks.

**Questions:**

My main question would be how the results could translate from basic fitting tasks to real-world learning problems, and how to best showcase this. In principle, I would've expected that this should be feasible to demonstrate with modern differentiable learning frameworks.

At the same time, I would like to highlight that this paper is pretty much out of my area. I tried my best to follow the derivations but it is not straightforward. In comparison to similar papers out of this area, the presentation makes this also difficult in addition to my lack of background knowledge in this space. From a pure paper presentation standpoint, this could be improved.

---

### Official Review · Reviewer_3qki · 2025-11-05

**Soundness:** 3
**Presentation:** 3
**Contribution:** 3
**Rating:** 6
**Confidence:** 3

**Summary:**

This is a comprehensive and well-structured paper introducing Simplex-FEM Networks. The authors have effectively presented a novel, theoretically grounded, and empirically strong alternative to standard MLPs and recent models like KANs, particularly excelling in areas requiring explicit geometry and sharp interfaces.

**Strengths:**

- The core idea is rooted in classical Finite Element Methods (FEM), providing a solid foundation with an established approximation rate. This theoretical backing is a significant advantage over many heuristic neural network architectures.

- The use of $C^r$ continuity constraints (specifically $C^0$ and partial $C^1$) allows explicit control over the smoothness of the predicted function, which is crucial for approximating functions with sharp features or discontinuities.

- SiFEN consistently matches or surpasses MLPs and KANs at matched parameter budgets. Notably, it shows marked improvement in calibration (lower ECE/Brier scores). Training the mesh vertices, triangulation, and an optional geometric warp ($\Phi_\theta$) allows the model to adapt its approximation basis to the data distribution and function complexity.

**Weaknesses:**

- The paper clearly acknowledges that mesh complexity scales poorly in high dimensions (the theoretical rate $M^{-m/d}$ gets very small for large $d$ unless $M$ is enormous). While the warp $\Phi_\theta$ is proposed as a mitigation strategy (flattening curvature or reducing effective dimension), the effectiveness of this for extremely high dimensions ($d \gg 50$) remains a primary practical concern.

- The performance is highly dependent on effective regularization for shape regularity and coverage. Degenerate simplexes and poor vertex initialization are explicit failure modes. The reliance on semi-discrete OT for coverage is sophisticated and may increase training complexity compared to simpler baselines.

- Table 17 breaks down the inference latency into "Point location" and "Local polynomial eval." The point location is currently implemented via kd-tree + walk or BVH. What are the computational challenges for the point location step specifically in high dimensions ($d=10, 50$)? Is there a computational limit on $M$ in high dimensions?

- The interpretability section focuses on the learned mesh structure (e.g., error vs. boundary crossings). The warp $\Phi_\theta$ is crucial to performance and is interpretable as a monotone triangular map. Can the authors offer a visual interpretation of the learned warp $\Phi_\theta$ on a 2D synthetic task?

**Questions:**

see weaknesses

---

### Note · Authors · 2025-11-13

I have read and agree with the venue's withdrawal policy on behalf of myself and my co-authors.